# Soy Isoflavones Protects Against Stroke by Inhibiting Keap1/NQO1/Nrf2/HO-1 Signaling Pathway: Network Pharmacology Analysis Combined with the Experimental Validation

**DOI:** 10.3390/ph18040548

**Published:** 2025-04-08

**Authors:** Huiming Xue, Zhen Feng, Chang Jin, Yue Zhang, Yongxing Ai, Jing Wang, Meizhu Zheng, Dongfang Shi

**Affiliations:** 1College of Life Sciences, Changchun Normal University, Changchun 130032, China; qx202310010@stu.ccsfu.edu.cn (H.X.); fengzhen0915@163.com (Z.F.); j1504408478@163.com (C.J.); 15942907284@163.com (Y.Z.); 2College of Animal Science, Jilin University, Changchun 130062, China; aiyx@jlu.edu.cn; 3Central Laboratory, Changchun Normal University, Changchun 130032, China; ccsfxy888@163.com

**Keywords:** stroke, soybean isoflavones, Nrf2, Keap1, rat

## Abstract

**Objectives**: Ischemic stroke is a severe neurological disorder with high morbidity, mortality, and disability rates, posing a substantial burden on patients, families, and healthcare systems. Soy isoflavone (SI), a naturally occurring phytoestrogen, has demonstrated promising neuroprotective effects. This study aimed to evaluate the anti-stroke efficacy of SI and elucidate its underlying mechanisms through integrated phytochemical profiling, network pharmacology, and both in vitro and in vivo experimental validation. **Methods**: Active constituents of SI were extracted via reflux and identified using liquid chromatography–mass spectrometry (LC-MS). Network pharmacology was employed to predict therapeutic targets and signaling pathways. The neuroprotective effects of SI were first assessed in PC12 cells subjected to oxygen–glucose deprivation/reoxygenation (OGD/R) injury in vitro. For in vivo evaluation, transient cerebral ischemia–reperfusion injury was induced using the bilateral common carotid artery occlusion (BCCAO) model in adult male ICR rats (27.3 ± 1.8 g; 6–8 weeks old), obtained from the Shanghai Experimental Animal Center, Chinese Academy of Sciences. Forty-eight rats were randomly assigned into four groups (*n* = 12): sham, model (BCCAO), SI-treated (100 mg/kg, oral gavage for 5 days), and edaravone (EDA)-treated (10 mg/kg, i.p., positive control). All procedures were approved by the Institutional Animal Care and Use Committee of Changchun Normal University (Approval No. 2024003, 13 March 2024) and conducted in accordance with the NIH guidelines and ARRIVE 2.0 reporting standards. **Results**: In vitro, SI significantly enhanced PC12 cell viability from 57.23 ± 2.88% to 80.76 ± 4.43% following OGD/R. It also reduced intracellular Ca^2+^ by 58.42%, lactate dehydrogenase (LDH) release by 37.67%, caspase-3 activity by 55.05%, and reactive oxygen species (ROS) levels by 74.13% (*p* < 0.05). A flow cytometry analysis revealed that OGD/R increased the apoptosis rate from 5.34% (control) to 30.85% (model group), which was significantly attenuated by SI treatment, especially in the 560 µg/mL group (20.00%), followed by the 140 and 280 µg/mL groups. In vivo, SI improved neurological scores from 8.3 ± 1.09 to 6.8 ± 1.68, reduced cerebral infarction volume by 18.49%, and alleviated brain edema by 10.42% (*p* < 0.05). SI also decreased malondialdehyde (MDA) and LDH levels by 31.15% and 39.46%, respectively, while increasing the activity of antioxidant enzymes: superoxide dismutase (SOD) by 11.70%, catalase (CAT) by 26.09%, and glutathione peroxidase (GSH-px) by 27.55% (*p* < 0.01). Scratch assay results showed that SI restored the impaired migratory ability of the OGD/R-treated PC12 cells, further supporting its role in cellular repair. A Western blot analysis demonstrated the upregulation of nuclear factor erythroid 2–related factor 2 (Nrf2), heme oxygenase-1 (HO-1), and NAD(P)H:quinone oxidoreductase 1 (NQO1) and the downregulation of Kelch-like, ECH-associated protein 1 (Keap1) in the cerebral ischemia–reperfusion model. **Conclusions**: These findings indicate that soy isoflavone confers significant neuroprotective effects against cerebral ischemia–reperfusion injury by enhancing endogenous antioxidant defense mechanisms, reducing oxidative stress, inhibiting apoptosis, and promoting cell migration. The protective effects are likely mediated through the activation of the Nrf2/Keap1 signaling pathway, supporting the therapeutic potential of SI in ischemic stroke treatment.

## 1. Introduction

Stroke is identified as a spectrum of disorders characterized by the interruption of blood circulation within the brain, leading to compromised neurological functions. These disorders are generally categorized into two principal types: hemorrhagic strokes and ischemic strokes (ISs). The latter, ischemic stroke, arises from the occlusion or constriction of arteries that are responsible for the cerebral blood supply, accounting for over seventy percent of all stroke instances [1,2]. Ischemic stroke is associated with a significant incidence of death, disability, and morbidity, posing a severe risk to patients’ lives and markedly affecting their quality of life [3]. Globally, stroke remains a leading cause of mortality and long-term disability, imposing substantial public health and economic burdens. Therefore, effective therapeutic strategies are urgently needed to mitigate its impact.

In the realm of physicochemical analysis, the development of ischemic stroke is predominantly attributed to mechanisms such as glutamate excitotoxicity, calcium accumulation, oxidative stress, apoptosis, and inflammation, among others [4,5,6,7]. The Keap1/Nrf2 signaling pathway is a key mechanism for maintaining the redox balance in the body [8]. Under normal physiological conditions, when cells are exposed to harmful external stimuli, such as environmental toxins or oxidative stress, reactive oxygen species (ROS) and electrophilic compounds can cause structural changes in Kelch-like, ECH-associated protein 1 (Keap1). This modification leads to the activation and nuclear translocation of a protein called nuclear factor erythroid 2–related factor 2 (Nrf2). Once inside the nucleus, Nrf2 functions as a transcription factor, turning on the expression of several protective genes that help the body defend itself against oxidative damage. These genes produce antioxidant and detoxifying enzymes, which play essential roles in neutralizing harmful molecules. Among these are enzymes like heme oxygenase-1 (HO-1) and NAD(P)H quinone dehydrogenase 1 (NQO1), which together help reduce oxidative stress and restore the cellular balance [9,10,11]. This signaling cascade is not only crucial for protection during acute injuries such as ischemic stroke but has also been implicated in the development and progression of a wide range of both cancerous and non-cancerous diseases [11]. Given its pivotal role in cellular defense, the Keap1/Nrf2 pathway has emerged as a promising therapeutic target for neuroprotection in ischemic stroke. However, effective modulators of this pathway remain limited, particularly among natural compounds.

Currently, the interest in natural medicinal research has surged globally. Evidence from numerous studies indicates a clear beneficial impact of botanicals and their bioactive constituents against ischemic stroke [12]. Soybeans, belonging to the legume family, have been identified as a significant source of isoflavones. Notably, soy isoflavones (SI), recognized as bioactive compounds, exhibit a range of biological activities. These include mimicking estrogen, combating aging and cancer, offering antioxidant and anti-inflammatory benefits, lowering lipid levels, preventing cardiovascular diseases, and improving memory functions [13,14]. Despite these diverse bioactivities, the neuroprotective potential of soy isoflavones in ischemic stroke is still underexplored. Their ability to modulate oxidative stress-related pathways, particularly Keap1/Nrf2, may provide a novel avenue for therapeutic development.

Network pharmacology integrates the principles of systems biology, omics technologies, and computational biology. This integration facilitates a comprehensive and synergistic exploration of the modes of action of pharmaceuticals, offering insights into their pharmacological effects [15]. Molecular docking techniques enable the prediction of protein–ligand interactions, including binding conformations and associated binding free energies, thereby supporting the functional validation of candidate compounds and clarifying their mechanisms of action [16]. The combined application of network pharmacology and molecular docking thus provides a powerful platform for investigating the multi-target and multi-pathway actions of complex natural products, such as soy isoflavones (SI), in ischemic stroke models.

Building on this integrated approach, the present study explored the potential role of SI in the prevention and treatment of ischemic stroke (IS), with a specific focus on redox regulation and oxidative stress modulation via the Nrf2 signaling pathway. An in vitro model of ischemic injury was established using Na_2_S_2_O_4_-induced hypoxic damage in PC12 cells, and a rat model of cerebral ischemia–reperfusion injury was induced via the embolization of the common carotid artery.

The primary objective of this study is to elucidate the neuroprotective mechanisms of soy isoflavones (SIs) in ischemic stroke (IS) by integrating in silico, in vitro, and in vivo approaches. First, network pharmacology and molecular docking were employed to systematically predict the putative targets and signaling pathways through which SI may exert therapeutic effects against IS. Next, a hypoxia-induced injury model in PC12 cells was established using sodium dithionite (Na_2_S_2_O_4_) to assess the in vitro protective effects of SI. Additionally, a rat model of cerebral ischemia–reperfusion injury was induced via common carotid artery embolization to evaluate the in vivo efficacy of SI. Particular attention was given to the Keap1/Nrf2/HO-1/NQO1 signaling axis to determine whether SI modulates oxidative stress through this pathway. By combining computational prediction with experimental validation, this study aims to provide mechanistic insights and preclinical evidence supporting the therapeutic potential of SI in ischemic stroke.

## 2. Results

### 2.1. Extraction and Identification of Soybean Isoflavone Cells: Experimental Results

The analysis and separation process for soybean isoflavones revealed a total of seven distinct isoflavone compounds, as demonstrated in Figure 1a. The mass spectral analysis and detailed data for each compound were systematically examined, with the findings presented in Figure 1b and Table 1. The identified isoflavone components included daidzin, glycitin, genistin, acetyldaidzin, acetylcitin, malonylgenistin, and acetylgenistin. The molecular structures of these seven isoflavones are depicted in Figure 1b.

### 2.2. Network Pharmacological Target Prediction

A review of the literature in the CNKI database revealed twelve components of soy isoflavone, comprising both three free glycosides and nine conjugated glycosides (refer to Table 2). Through the analysis of pharmacokinetic parameters using the SwissADME database, three free glycosides were identified as potent active ingredients: daidzein, genistein, and glycitein. In total, 111 targets for soy isoflavone were derived from two pharmacological databases (Figure 2a), while 4598 disease-related targets for IS were collated from four disease-specific databases (Figure 2b), leading to the identification of 84 common targets (Figure 2c). Utilizing String11.5, a PPI network was constructed, comprising 84 nodes and 524 connections (Figure 2d). Visualization using Cytoscape 3.8.0 indicated that nodes with a larger size and darker color within the figure represent a greater impact on IS. The top five pivotal genes, identified through the MCC algorithm in the cytoHubba plugin, were MMP9, EGFR, PPARG, PTGS2, and ESR1(Figure 2e). Afterwards, an “drug-component-disease-target” network diagram was constructed (Figure 2f).

The GO analysis forecast that the principal biological processes involve reactions to xenobiotic stimuli, the cyclooxygenase pathway, the positive regulation of the apoptotic process, the cellular response to reactive oxygen species, signal transduction, the inflammatory response, and antifolate resistance, among others. Regarding molecular functions, the key activities identified include enzyme binding, protein binding, iron ion binding, interaction with proteins and phosphatases, kinase activity, and ATP binding (Figure 2g).

The investigation into soy isoflavone’s role in mitigating IS via the KEGG pathway analysis unveiled its involvement across a spectrum of signaling pathways. These pathways encompass IL-17 associated with IS, along with the FoxO, cAMP, TNF, and ErbB signaling pathways and the regulation of TRP channels through inflammatory mediators. Additional pathways implicated include the Ras, neurotrophin, VEGF, and Relaxin signaling mechanisms (Figure 2h). Combined with a literature review, we hypothesized that soy isoflavone may delay the translation of brain tissue damage after a stroke by affecting redox-related signaling pathways. For the in vitro and in vivo experimental verification, we chose the classical keap1/NQO1/Nrf2/HO-1 signaling pathway.

Figure 2i,j categorized the top 50 pathways enriched by KEGG, and these pathways could be classified into four categories, including metabolism (amino acid metabolism, lipid metabolism, global and overview maps, and energy metabolism), cellular processes (transport and catabolism, cell growth and death), organismal systems (immune system, endocrine system, sensory and regeneration, nervous system, development, and regeneration), human diseases (neurodegenerative diseases, cardiovascular diseases), signal transduction (Sphingolipid signaling pathway, FoxO signaling pathway, MAPK signaling pathway, ErbB signaling pathway, VEGF signaling pathway, PI3K Akt signaling pathway, Ras signaling pathway, and so on).

Based on this, soy isoflavones’ active ingredients (daidzein, genistein, and glycitein) were molecularly docked with the pathway protein Nrf2/keap1/NQO1/HO-1 (Figure 3a). An affinity of less than −4.25 kcal·mol^−1^ indicates binding activity between the ligand and the target; an affinity of less than −5.0 kcal·mol^−1^ indicates good binding activity; and an affinity of less than −7.0 kcal·mol^−1^ indicates strong docking activity. Their affinities are shown in Figure 3b.

### 2.3. Cell Experimental

#### 2.3.1. Establishment of Soy Isoflavone Concentration

The MTT assay was used to assess the cytotoxicity of soy isoflavones and determine their concentration. The results indicated that as the concentration of soy isoflavones increased, their cytotoxicity was enhanced, particularly when concentrations exceeded 2240 μg/mL, which significantly reduced cell viability (*p* < 0.05). Therefore, concentrations below 2240 μg/mL were considered within the biosafe range. Based on this, concentrations of 140 μg/mL, 280 μg/mL, and 560 μg/mL were selected for subsequent experiments. In further experiments, the concentration range of soy isoflavones was expanded up to 40 g/L, and the EC_50_ value was calculated. The EC_50_ of soy isoflavones was determined to be 37.01 g/L (Figure 4).

#### 2.3.2. Morphological Changes in PC12 Cells

A microscopic examination revealed that the normal control group exhibited a higher quantity of PC12 cells and superior cell proliferation. In contrast, the model group showed a reduction in both the PC12 cell count and synaptic connections. The groups treated with soy isoflavone and the positive control showed an increased number of viable cells, a decrease in cellular damage, and a partial restoration of cells to their normal morphology (Figure 5a).

#### 2.3.3. The Effect of the Treatment on the PC12 Cells with Increasing Concentrations of Soy Isoflavone

Figure 5b presents a comparative study showing that PC12 cells in the model group displayed a significantly lower survival rate compared to the control group. However, treating these cells with soy isoflavones at a dose of 560 µg/mL markedly improved their survival rates. The data presented in Figure 5c–f elucidate that the levels of Ca^2+^, ROS, LDH, and Caspase-3 were considerably elevated in the model group when juxtaposed with the control group. Remarkably, the administration of soy isoflavone at a 560 µg/mL dosage markedly diminished these levels, indicating a protective effect against cellular stress and apoptosis. The flow cytometry analysis, as shown in Figure 5g,h, revealed an apoptosis rate of 5.34% in the control group, which surged to 30.85% in the model group, underscoring the detrimental impact of the OGD/R treatment. Conversely, the soy isoflavone-treated groups displayed a significant reduction in apoptosis rates, with the 560 µg/mL dosage group showing the most pronounced decrease to 20.00%, followed by the 140 µg/mL and 280 µg/mL dosage groups. The edaravone-positive control also showed a decrease in apoptosis, validating the protective efficacy of both treatments against cellular apoptosis induced by OGD/R. Furthermore, Figure 5i,j highlight the detrimental effect of OGD/R treatment on the migratory capacity of PC12 cells. The scratch assay results indicate a substantial recovery in the scratch area for the groups treated with soy isoflavone and the positive control, evidencing the compound’s role in facilitating cellular repair and migration post-injury.

#### 2.3.4. Effect of Soybean Isoflavones on Nrf2 Pathway in Sodium Metabisulfite-Treated PC12 Cells

To explore the protective mechanisms of soy isoflavones, the study utilized Western blotting to assess the nuclear Nrf2 expression levels and the expressions of Keap1, HO-1, and NQO1. As depicted in Figure 6, the soy isoflavone and edaravone treatments significantly elevated the nuclear expression of Nrf2 when compared to the model group (*p* < 0.01). Furthermore, soy isoflavones distinctly reduced the expression of Keap1 while simultaneously increasing the expressions of HO-1 and NQO1 (*p* < 0.01 for each). These findings suggest that the protective effects of soy isoflavones may be intricately linked to the modulation of the Nrf2 and Keap1 pathways (Figure 6).

### 2.4. Animal Experimentation

#### 2.4.1. Soybean Isoflavones Improved the Neurological Function Score, Brain Tissue Water Content, and Cerebral Infarct Volume of Rats in Each Group

In order to evaluate the therapeutic impact of soy isoflavones on cerebral ischemic injury, this study conducted an analysis focusing on the neurological function score, the moisture content of brain tissue, and the extent of cerebral infarction following ischemic damage. The investigation revealed that, in comparison to the sham-operated control group, the model group exhibited significantly elevated scores for behavioral dysfunction, increased moisture levels in brain tissue, and expanded cerebral infarction volumes, highlighting extensive neuronal injury. In contrast, pretreatment with soy isoflavones (100 mg/kg) administered via intraperitoneal injection notably reduced the scores for neurological dysfunction, decreased the water content within the brain, and minimized the volume of cerebral infarction (Figure 7). Compared to the model group, the application of soy isoflavones and edaravone markedly enhanced recovery from neurological dysfunction, with statistical significance (*p* < 0.01 for both treatments). These significant findings are systematically presented in Table 3.

#### 2.4.2. Soybean Isoflavones Improved the Brain Histopathology of the Rats in Each Group

Following HE staining and microscopic examination, the sham-operated group exhibited no signs of ischemic alterations. Observations revealed that the architecture and cytoplasm of neuronal cells remained intact, with nucleoli distinctly visible, and no cellular abnormalities were observed near or within the vasculature. Contrastingly, in the model group, focal ischemia–reperfusion injuries were predominantly localized to the brain’s central region, characterized by loosened brain tissue, edema, and neuronal tract ischemic changes, including pyknosis within the affected lesions. Soy isoflavone pretreatment markedly mitigated the neuronal damage within the ischemic zone, as evidenced in Figure 8.

#### 2.4.3. Effects of Soybean Isoflavones on Activities of SOD, GSH-px, CAT, and Contents of MDA and LDH in Rat Brains

Compared with the control group subjected to a sham procedure, the model group displayed diminished activity levels of SOD, GSH-px, and CAT within the cerebral tissue. This group also showed elevated levels of MDA and LDH, with all the differences being statistically significant (*p* < 0.01). Following treatment with soybean isoflavones and edaravone, a significant reduction in MDA concentrations and LDH release was observed. Concurrently, there was a significant enhancement of the activities of SOD, GSH-px, and CAT (*p* < 0.01), as detailed in Table 4.

#### 2.4.4. Soybean Isoflavones Reduce Oxidative Stress Injury by Increasing Nrf2 Signaling Pathway

Figure 9 illustrates that, in comparison to the model group, treatment with soybean isoflavones and edaravone significantly augmented the nuclear expression of Nrf2 (*p* < 0.01). Additionally, soybean isoflavones markedly reduced the expression of Keap1 while simultaneously enhancing the expression levels of HO-1 and NQO1 (*p* < 0.01 for all). These findings suggest a strong correlation between the protective effects of soybean isoflavones and the regulatory activities on the Nrf2 and Keap1 pathways.

## 3. Discussion

Ischemic stroke has emerged as a critical health issue worldwide. The primary treatment options available are thrombolytic drugs, which, despite their efficacy, pose significant risks to health. These drugs are only effective within a narrow therapeutic window of 3 to 6 h following a stroke; administration beyond this period risks causing ischemia–reperfusion injury [17]. Studies have elucidated that the cessation of blood flow results in ischemia, and the subsequent restoration of circulation, or reperfusion, leads to an influx of oxygen, generating excessive oxygen free radicals [18]. Concurrently, this process suppresses the activity of antioxidant enzymes, precipitating oxidative stress that exacerbates brain damage. Although thrombolytic agents offer crucial benefits for acute ischemic stroke, potentially saving lives, their long-term application is hindered by their chemical composition, which can interfere with patient metabolism. On the other hand, natural biological agents present a more favorable profile, characterized by gentler effects and reduced toxicity and side effects, making them more suitable for extended use [19].

Soybean isoflavones possess multiple pharmacological effects, significantly contributing to antioxidation, mitigating inflammation following cerebral ischemia, cholesterol reduction, and the prevention of cardiovascular diseases [20,21]. This study utilized the reflux method to extract isoflavones from soybean, with the constituents of the soybean extract subsequently identified through HPLC-DAD-MS. The primary components identified were daidzin, glycitin, genistin, acetyldaidzin, acetylcitin, malonylgenistin, and acetylgenistin. The network pharmacology analysis suggested that soy isoflavones may modulate the oxidative stress in brain tissues induced by stroke through their influence on redox signaling pathways, thereby offering potential prevention and treatment for ischemic stroke. In vivo experiments entailed establishing a rat model of transient global cerebral ischemia–reperfusion by occluding the bilateral common carotid arteries. Preliminary findings revealed that soybean isoflavone pretreatment markedly ameliorated neurological damages in cerebral ischemia–reperfusion rats, significantly diminishing the brain water content and the infarct volume. Additionally, there was a significant elevation in the activities of SOD, GSH, and CAT within the brain tissues, alongside a marked reduction in MDA and LDH levels. Morphological observations further confirmed the significant mitigation of cerebral lesions by soybean isoflavones in rats subjected to cerebral ischemia–reperfusion injury.

An imbalance between the generation of ROS and the capacity of the organism to counteract their negative effects with antioxidants leads to oxidative stress [22]. The body possesses an elaborate defense system against free radicals, comprising antioxidant enzymes such as SOD, CAT, and GSH-px [23]. SOD, a metalloproteinase found across various tissues, cells, and organelles, plays a crucial role in scavenging excess free radicals, thus maintaining the cellular peroxidation balance. Its activity is often used as an indicator of the body’s antioxidant capability [24]. Furthermore, GSH-px and CAT effectively catalyze the breakdown of hydrogen peroxide, eliminating harmful superoxide metabolites to reduce peroxidative damage [25]. Therefore, the body’s antioxidant capacity can be gauged by the activities of SOD, GSH-px, and CAT and the levels of ROS. Ischemia/reperfusion injury induces a surge in ROS, leading to an increase in lipid peroxides, which are then converted into MDA (malondialdehyde) [26]. Simultaneously, the activity of Ca^2+^ pumps declines, causing an elevation in cellular Ca^2+^ concentration and accumulation, which interferes with Ca^2+^ signal transduction [27]. An increase in Ca^2+^ concentration triggers the production of free fatty acids, leading to a surge in free radicals [28], which disturbs both the energy and the intracellular environmental equilibrium. This process activates Caspase-3, fostering neuronal apoptosis, exacerbating cellular damage, and heightening the risk of myocardial infarction. Concurrently, oxidative stress can escalate serum LDH (lactate dehydrogenase) levels, causing further cellular and tissue damage. Thus, serum concentrations of MDA, Ca^2+^, Caspase-3, and LDH serve as indicators of oxidative harm. It was discovered that SI significantly mitigates the damage to PC12 cells induced by oxygen–glucose deprivation/reoxygenation, reduces apoptosis, boosts cell viability, and diminishes levels of Ca^2+^, LDH, and ROS. Hence, SI exerts a notable protective effect against Na2S2O4-induced injury in PC12 cells, likely through mitigating oxidative stress. Additionally, this investigation reveals that pretreatment with soybean isoflavones markedly lowers MDA and LDH levels in the brain tissue of rats suffering from cerebral ischemia–reperfusion injury while notably enhancing the activities of SOD, GSH-px, and CAT. Consequently, soybean isoflavones effectively elevate antioxidant enzyme activity in rats subjected to cerebral ischemia–reperfusion injury, thereby alleviating oxidative stress-induced damage.

Nrf2 is recognized as a key defender against oxidative stress in ischemic stroke [29,30]. Research indicates that Nrf2 not only indirectly modulates the expression of various transcription factors but also contributes to neuroprotection by dampening inflammatory responses, combating oxidative stress, diminishing apoptosis and necrosis, and fostering cell growth and differentiation [31]. Moreover, the Nrf2 signaling pathway plays a crucial role in neuroprotection, as evidenced by studies on numerous traditional Chinese medicinal compounds [32]. Normally, Nrf2 forms an inactive complex with Keap1 [33]. Upon activation, Nrf2 separates from Keap1, migrates from the cytoplasm to the nucleus, and binds to the ARE, thereby activating antioxidant enzymes like HO-1 and NQO1 [34]. This investigation highlighted that soy isoflavones notably increase the concentrations of Nrf2, HO-1, and NQO1, while significantly reducing Keap1 levels (Figure 10). It suggests that the neuroprotective mechanisms of soy isoflavones could be linked to stimulating the Nrf2 signaling pathway. Through this pathway, the regulation of antioxidant responses to oxidative stress from cerebral ischemia–reperfusion is achieved, potentially facilitating cellular growth, differentiation, and angiogenesis, thus contributing to neuroprotection.

Research indicates that soy isoflavones exhibit a protective capacity in the context of cerebral ischemia–reperfusion injuries. This protective effect is attributed to the enhancement of antioxidant enzyme activities and the suppression of oxidative stress. Furthermore, soy isoflavones’ defense against ischemic stroke is linked to the stimulation of Nrf2 and the reduction of Keap1 activity. These findings are consistent with previous in vivo and in vitro studies, further supporting the multi-target neuroprotective potential of SI.

In vitro, Zhang et al. (2022) reported that SI protected PC12 cells from CoCl_2_-induced hypoxic injury by activating Nrf2 and suppressing the p38 MAPK and AKT–mTOR pathways, thereby reducing ROS levels, enhancing cell viability, and inhibiting apoptosis [35]. These mechanisms align with our observations in the OGD/R model, where SI demonstrated antioxidant and anti-apoptotic effects. In vivo, Schreihofer et al. (2005) found that a high-soy diet significantly reduced the infarct size in female rats subjected to permanent middle cerebral artery occlusion (MCAO), indicating notable neuroprotective effects under physiological conditions [36]. Similarly, Huang et al. (2009) showed that SI alleviated brain tissue damage and improved neurological function in ischemic rats by modulating the Notch signaling pathway [37]. Moreover, Li et al. (2025) demonstrated that genistein, a major component of SI, mitigated cerebral ischemia–reperfusion injury in rats by inhibiting the Wnt/Ca^2+^ signaling pathway, emphasizing its role in calcium homeostasis [38]. These studies reinforce our findings that SI exerts neuroprotective effects through multiple signaling pathways, particularly those related to oxidative stress and apoptosis regulation.

In conclusion, our findings are in strong agreement with existing in vivo and in vitro research, underscoring the multifaceted mechanisms by which soy isoflavones alleviate ischemic brain injury. These results support the therapeutic potential of SI and provide a solid foundation for its future clinical application. Although this study provides evidence supporting the neuroprotective role of soy isoflavones through the activation of the Nrf2 signaling pathway, several limitations should be noted. First, the study was conducted using a single animal model (BCCAO) and PC12 cells, which may not fully reflect the complexity of human ischemic stroke pathology. Second, the exact active monomer(s) within the soy isoflavone extract responsible for the observed effects remain to be identified. Third, the long-term safety and efficacy of soy isoflavones were not assessed. Future studies should explore the effects of individual isoflavone components, investigate additional models of stroke (e.g., MCAO), and assess chronic outcomes such as cognitive function and neuroregeneration.

## 4. Materials and Methods

### 4.1. Chemical, Reagents, and Software

Soy isoflavones standards, including Daidzin, Glycitin, Genistein, Acetyldaidzin, Acetylglycitin, Malonylgenistein, and Acetylgenistein, were purchased from Aladdin Industrial Corporation (Cat# various; Shanghai, China). Acetonitrile (HPLC grade) was obtained from Thermo Fisher Scientific (Cat# A955-4; Waltham, MA, USA). Ultrapure water was prepared using a Milli-Q system (Millipore, Billerica, MA, USA). Analytical-grade reagents were provided by Beijing Chemical Works (Beijing, China). Kits for superoxide dismutase (SOD), glutathione peroxidase (GSH-px), malondialdehyde (MDA), and total protein were from the Nanjing Jiancheng Bioengineering Institute (Cat# A001-3, A005, A003-1, A045-2-2; Nanjing, Jiangsu, China). Kits for catalase (CAT), 3-(4,5-dimethylthiazol-2-yl)-2,5-diphenyltetrazolium bromide (MTT), lactate dehydrogenase (LDH), reactive oxygen species (ROS), Fluo-3 acetoxymethyl ester (Fluo-3AM), and caspase-3 were obtained from Beyotime Biotechnology (Shanghai, China). Other chemicals, including internal standard, Earle’s Balanced Salt Solution (EBSS), phosphate-buffered saline (PBS), sodium dithionite (Na_2_S_2_O_4_), and edaravone, were purchased from Sigma-Aldrich (St. Louis, MO, USA).

Primary antibodies: Anti-Nrf2 (Cat# 12721, Cell Signaling Technology, Danvers, MA, USA); Anti-Keap1 (Cat# ab227828), Anti-HO-1 (Cat# ab68477), Anti-NQO1 (Cat# ab34173), and Anti-β-actin (Cat# ab8227) (all from Abcam, Cambridge, UK). The GraphPad Prism version 9.5.1 (GraphPad Software, San Diego, CA, USA) was used for the statistical analysis. ImageJ version 1.53 (National Institutes of Health, Bethesda, MD, USA) was used for the densitometric analysis. FlowJo version 10.8.1 (BD Biosciences, Ashland, OR, USA) was used for the flow cytometry data analysis. Cytoscape version 3.9.1 (The Cytoscape Consortium, San Diego, CA, USA), AutoDockTools version 1.5.7, AutoDock Vina version 1.2.3, and PyMOL version 2.5.4 (Schrödinger, LLC, New York, NY, USA) were used for molecular docking and visualization.

### 4.2. Extraction and Identification of Soybean Isoflavones

A precise 10 g sample of soybean was measured and subjected to a reflux extraction process using 100 mL of 65% ethanol, conducted twice for a duration of 2 h each. The resultant filtrate was then concentrated and pooled together, followed by dilution to a fixed volume of 10 mL with methanol. Subsequent to comprehensive homogenization, the filtration of the solution through a membrane with a pore size of 0.45 μm was performed in preparation for the analytical procedures. The identification of chemical components within the soybean extracts was carried out utilizing HPLC-DAD-MS [39].

The procedure for high-performance liquid chromatography (HPLC) involved using a binary linear gradient elution technique [40]. Here, acetonitrile served as mobile phase A, while mobile phase B consisted of water containing 0.05% phosphoric acid. The gradient schedule for the mobile phases was meticulously programmed: starting with mobile phase B at 90%, it was gradually reduced to 82% over the first 10 min, then further decreased to 80% between 10 and 13 min, followed by a drop to 60% from 13 to 30 min, and was ultimately adjusted to 50% for the final 10 min, from 30 to 40 min. The flow rate was consistently held at 1.0 mL/min, with a sample injection volume of 10 μL, and the detection was conducted at a wavelength of 254 nm, all while maintaining the column at a stable temperature of 30 °C.

For the mass spectrometry, the diode array detector (DAD) was linked to the mass spectrometer via a three-way valve, employing an electrospray ionization (ESI) source in positive ion mode. The scan range was set between M/Z 50 and 1000. The ion source spray voltage was established at 4.5 kV, with a sheath gas flow rate of nitrogen set at 20 L/min and an ion trap pressure at 3.1 × 10^7^ Pa. The temperature for the metal capillary was 250 °C with a voltage of 20 V. The resolution for ESI-MS^n was maintained at 1.0 Da, and the collision-induced dissociation (CID) collision energy ranged from 20 to 30%.

### 4.3. Network Pharmacology Research

The research on soy isoflavone was initiated by exploring the CNKI database using “soy isoflavone” as the search term to compile a list of soy isoflavone components. The SwissADME online server was utilized to screen soy isoflavone’s active ingredients, evaluating each component’s pharmacokinetic parameters and adherence to the Lipinski Rule of Five [15]. The potential targets of these active ingredients were then predicted through the Swiss Target Prediction and PharmMapper databases. To identify relevant targets for ischemic stroke, searches were conducted in the GeneCards, OMIM, PharmGkb, and DisGeNET databases. The identified soy isoflavone targets were aligned with those related to drug actions to pinpoint common targets, and a Venn diagram was created to visualize these intersections. Subsequently, these intersecting targets were analyzed for Protein–Protein Interactions (PPIs) using the String database (version 12.0), with the findings visualized in Cytoscape 3.8.0. The intersection targets underwent further analysis of the GO and KEGG pathways using the DAVID website. The top 50 KEGG pathways were classified and summarized according to the first 6 classifications in the KEGG pathway database. Molecular docking between the core components and the core targets was performed based on the above analysis. The 3D crystal structures of the target proteins were obtained from the Protein Data Bank (PDB). Their pdb format files were downloaded, de-watered, and hydrogenated in AutoDockTools (version: 1.5.7) and then selected as receptors and saved as pdbqt files. The mol2 files of the active components were obtained through the TCMSP database, hydrogenated in AutoDockTools, selected as ligands, and exported as pdbqt files. Molecular docking was carried out through AutoDock Vina (version: 1.2.3) and the affinity of each component for the target was obtained and visualized by PyMOL (version: 2.7.0) and the ZBH-Center for Bioinformatics. The results were illustrated through histograms and bubble maps representing the pathway information, which were crafted by the bioinformatics.com.cn platform, accessed last on 28 December 2023 [41].

### 4.4. In Vitro Studies

#### 4.4.1. PC12 Cell Culture

PC12 cells, acquired from the Chinese Academy of Sciences in Shanghai, were cultivated in a nutrient-rich Dulbecco’s Modified Eagle Medium (DMEM), a high-glucose medium, which was further supplemented with 10% fetal bovine serum and a 1% mixture of penicillin and streptomycin antibiotics. These cells were then incubated at a constant temperature of 37 °C in an environment enriched with 5% CO_2_ for a duration spanning between three and four d, until they reached a confluence of 70–80%. Following this phase of growth, a solution containing 0.25% trypsin was employed to gently detach the cells, facilitating their transfer to new culture containers for continued propagation. In preparation for a range of experimental activities, the cells were subsequently allocated into 96-well plates, with 100 μL of the cell culture introduced into each well. Post-seeding, an additional period of incubation was undertaken in a 5% CO_2_ atmosphere at 37 °C, extending from one to two d, to allow the cells to achieve a confluence between 80 and 90%, thereby rendering them suitable for further experimental investigations [42].

#### 4.4.2. Establishment of Soy Isoflavone Concentrations

PC12 cells were inoculated into 96-well culture plates at a density of 9 × 10^4^/mL, 100 μL per well, and incubated at 37 °C with 5% CO_2_ for 24 h. Afterwards, the original culture medium was discarded, and different concentrations of soy isoflavones (140, 280, 560, 1120, 2240, 4480, 8960, 17,920, and 35,840 μg/mL) were added to incubate the cells for 24 h. At the end of the incubation period, the cytotoxicity of the PC12 cells was detected by MTT.

#### 4.4.3. The Establishment and Experimental Grouping of a Glucose–Oxygen Deprivation/Reoxygenation (OGD/R) Ex Vivo Stroke Model

To evaluate the protective impact of a natural isoflavone, SI, on PC12 cells subjected to OGD/R-induced injury, the cells, prepared as previously mentioned, were allocated to various groups: the control group received no OGD/R injury treatment. In the model group, the cells were treated with 100 µL of glucose-free EBSS solution enriched with 10 mmol/L Na_2_S_2_O_4_ for 2 h. Subsequently, the medium was replaced with a standard culture medium to allow reoxygenation, and the cells were further incubated for 24 h, establishing an OGD/R injury model to mimic ischemic brain damage. For the treatment groups, OGD/R conditions were amended with a standard culture medium containing different concentrations of SI (1.56 µM, 3.13 µM, and 6.25 µM) for 24 h. Additionally, a positive control group was treated with 10 µM edaravone under OGD/R conditions (*n* = 6), to compare the efficacy of SI against a known neuroprotective agent [43].

#### 4.4.4. Morphological Observation

Following the incorporation of various treatment agents, the PC12 cells were maintained at 37 °C for a 24 h period. Post-treatment, the cells underwent examination under an inverted microscope to assess any morphological alterations across the specified groups (*n* = 6).

#### 4.4.5. MTT Method to Measure Cellular Activity

Following treatment, each well in a 96-well cell culture plate was treated with 10 µL of 3-(4,5-dimethylthiazol-2-yl)-2,5-diphenyltetrazolium bromide (MTT) solution (5 g/L concentration) in a light-protected environment. After incubating for 4 h at 37 °C in a 5% CO_2_ atmosphere, the supernatant was discarded. Subsequently, to dissolve the resultant formazan crystals, 100 µL of dimethyl sulfoxide (DMSO) was added to each well. The optical density of the solutions in each well was measured at 490 nm wavelength using an enzyme-linked immunosorbent assay, with measurements taken within 10 to 15 min. All the data are presented as the mean ± SD (*n* = 6 per group), and statistical comparisons were performed using a one-way ANOVA followed by Tukey’s post hoc test.

#### 4.4.6. Detection of Ca^2+^ Content in PC12 Cells

After applying the treatment, the cells were incubated for one hour at 37 °C in a 5% CO_2_ atmosphere with the culture medium that included the Fluo-3AM fluorescent probe at a concentration of 2.5 µmol/L. Following this incubation period, any residual probe was meticulously washed away, and the fluorescence indicative of intracellular Ca^2+^ levels was detected using a fluorescence microscope [44]. The intensity of this fluorescence, with the excitation and emission wavelengths set at 488 nm and 530 nm, respectively, was quantitatively measured through enzyme-linked assay techniques. To account for differences in cell density, an MTT assay was utilized to standardize the cell density before calculating the concentrations of intracellular Ca^2+^ in PC12 cells.(1)$$H=\frac{A}{O}÷B\times 100\%$$

In this methodology, *H* represents the content of Ca^2+^ within the sample, *A* denotes the value obtained from the calcium ion assay for each individual well, and *O* signifies the absorbance measured for each specific well. Additionally, MTT *B* refers to the average value derived from the MTT assay across all the wells within a given treatment group, with each group comprising six replicates. All the data are presented as the mean ± SD (*n* = 6 per group), and statistical comparisons were performed using a one-way ANOVA followed by Tukey’s post hoc test.

#### 4.4.7. LDH Release Assay in PC12 Cells

Following the completion of the drug treatment, the release of lactate dehydrogenase (LDH) from the PC12 cells was quantitatively assessed as per the protocol provided by the LDH assay kit, with the experiment being replicated six times [45]. All the data are presented as the mean ± SD (*n* = 6 per group), and statistical comparisons were performed using a one-way ANOVA followed by Tukey’s post hoc test.

#### 4.4.8. Detection of ROS Levels in PC12 Cells

Following the treatment, the PC12 cells were exposed to a serum-free culture medium containing a 10 μmol/L DCFH-DA probe for 20 min in an environment of 5% CO_2_ at 37 °C. Subsequently, the cells were washed to remove any residual serum-free medium. The levels of ROS within the PC12 cells were then quantified utilizing an enzyme marker, employing the same calculation method as previously described for Ca^2+^ levels, with the process repeated in six samples [46]. All the data are presented as the mean ± SD (*n* = 6 per group), and statistical comparisons were performed using a one-way ANOVA followed by Tukey’s post hoc test.

#### 4.4.9. Caspase-3 Release Assay in PC12 Cells

Following the completion of the drug treatment, the release of Caspase-3 from the PC12 cells was quantified in accordance with the guidelines provided by the Caspase-3 assay kit, with the analysis conducted on six replicates [47]. All the data are presented as the mean ± SD (*n* = 6 per group), and statistical comparisons were performed using a one-way ANOVA followed by Tukey’s post hoc test.

#### 4.4.10. Apoptosis Assay

Following the administration of the drug, PC12 cell apoptosis was evaluated using flow cytometry with the assistance of the Annexin V-FITC Apoptosis Assay Kit (Beyotime Biotechnology, Shanghai, China). This kit employs fluorescein isothiocyanate-labeled Annexin V, a phospholipid-binding protein, in conjunction with propidium iodide (PI) for staining. The apoptotic rates were determined using the FACSCanto II flow cytometry system(BD Biosciences, San Jose, CA, USA), and the proportion of apoptotic cells was analyzed and calculated using FlowJo software version 10.6.2, across three experimental replicates. All the data are presented as the mean ± SD (*n* = 3 per group), and statistical comparisons were performed using a one-way ANOVA followed by Tukey’s post hoc test.

#### 4.4.11. Wound Healing Assay

For the cell scratch assay, a 6-well plate was prepared. Parallel lines spaced 0.5 cm apart were marked on the underside of the plate using a marker, with three lines per well to delineate the observation area. The cells were plated at a concentration of 5 × 10^5^ cells per milliliter in each well. Following this, a pipette tip was utilized to generate a straight scratch through the cell monolayer. Any detached cells resulting from the scratch were removed by washing with PBS. Images of the scratch were captured at 0 and 24 h post-scratch using a microscope to evaluate cell migration. The assessment was based on measuring the changes in the width of the scratch, indicating the cells’ migratory capacity. All the data are presented as the mean ± SD (*n* = 3 per group), and statistical comparisons were performed using a one-way ANOVA followed by Tukey’s post hoc test.

#### 4.4.12. Western Blot Analysis

For protein sample preparation, the cells underwent lysis on ice for approximately 20 min. This was followed by centrifugation at 4 °C at 13,000× *g* for 5 min. The protein concentrations in the resulting supernatants were accurately measured using the bicinchoninic acid (BCA) protein assay method. Post-separation, the proteins were subjected to a 5–15% sodium dodecyl sulfate–polyacrylamide gel electrophoresis (SDS-PAGE) electrophoresis process, subsequently being transferred to polyvinylidene fluoride (PVDF) membranes. To inhibit non-specific binding, these membranes underwent incubation with 5% skimmed milk at room temperature for an hour. Overnight incubation at 4 °C was carried out with primary antibodies specifically targeting Nrf2, Keap1, HO-1, β-actin, and NQO1, with dilutions prepared at 1:1000 for all but NQO1, which was diluted to 1:4000. Following thorough washes, the membranes were treated with secondary antibodies diluted to 1:1000 for an hour at room temperature. After additional washing steps, the protein bands were detected with an ECL detection system and captured for analysis. A semi-quantitative assessment of these bands was performed using ImageJ (version 1.53) software, providing an analytical basis for evaluating protein expression levels. All the data are presented as the mean ± SD (*n* = 3 per group), and statistical comparisons were performed using a one-way ANOVA followed by Tukey’s post hoc test.

### 4.5. In Vivo Studies

#### 4.5.1. Animals and Treatment

Forty-eight adult male ICR rats (average body weight 27.3 ± 1.8 g, age 6–8 weeks) were obtained from the Shanghai Experimental Animal Center, Chinese Academy of Sciences (Shanghai, China). The animal procedures were approved by the Institutional Animal Care and Use Committee (IACUC) of Changchun Normal University (Approval No. 2024003, approved on 13 March 2024). All animal handling followed the guidelines set by the National Institutes of Health (NIHs) and complied with the ARRIVE 2.0 reporting standards.

#### 4.5.2. Housing Conditions and Welfare Measures

Rats were housed in ventilated cages under a controlled environment (temperature: 18–22 °C; relative humidity: 50–60%; 12/12 h light/dark cycle), with ad libitum access to standard chow and water. Daily monitoring was performed for food intake, weight, and clinical symptoms. All efforts were made to minimize animal suffering, and the minimum number of animals required for statistical validity was used.

#### 4.5.3. Experimental Design and Surgical Procedure

A total of 48 rats (body weight 25–30 g) were randomly assigned into four groups (*n* = 12 per group) using a random number generator: The sham group underwent identical surgical procedures without carotid occlusion. The model group received bilateral common carotid artery occlusion (BCCAO) to induce cerebral ischemia–reperfusion injury. The SI treatment group was treated with soy isoflavones (100 mg/kg, dissolved in saline) via intragastric administration for 5 consecutive d prior to ischemia [48,49]. The edaravone group received edaravone (10 mg/kg, i.p.), a standard neuroprotective agent, once daily for 5 d as a positive control. 

Transient cerebral ischemia–reperfusion injury was induced using the bilateral common carotid artery occlusion (BCCAO) model [50]. The rats were anesthetized with pentobarbital sodium (45 mg/kg, i.p.). A midline neck incision (~3 cm) was made to isolate and occlude the bilateral common carotid arteries with vascular clips for 60 min, followed by reperfusion for 24 h. For the sham group, the arteries were exposed but not occluded. Intraoperatively, the rectal temperature of the rats was maintained at 37.0 ± 0.5 °C using an automated thermostatic heating pad. Postoperatively, the animals were housed in a climate-controlled environment (ambient temperature: 23 ± 1 °C; relative humidity: 55 ± 5%) with sterile bedding and ad libitum access to food/water. Subsequently, behavioral tests were performed on the animals. After the completion of the behavioral tests, the animals were euthanized under deep anesthesia by cervical dislocation, in accordance with the AVMA guidelines. Brain tissues were immediately harvested and processed as follows: 5 rats per group for histopathological examination, 3 rats per group for Western blot analysis, and 4 rats per group for HE staining. All the relevant pairwise comparisons among the four experimental groups (control, model, SI, and edaravone) were pre-defined and included in the statistical analysis to comprehensively assess the treatment effects and control efficacy.

#### 4.5.4. Neurological Function Score

Neurological function was assessed according to the method described by Bederson et al. The assessment was conducted in a blinded manner to exclude an observer bias. Briefly, each rat was initially placed on a flat surface to observe its spontaneous movements. Circling behavior toward the side opposite the ischemic region was scored as 1 point. When the rats were gently lifted by the tail, a flexion of the elbow was scored as 2 points, an internal rotation of the shoulder as 3 points, and a wrist flexion of the forelimb as 1 point. Additionally, pushing resistance tests were performed, with scores from 1 to 3 assigned based on the degree of motor deficit (greater distances corresponded to higher scores). The rats were placed on a metal mesh and were further assessed for contralateral muscle tension reduction, with the severity scored from 1 to 3 points according to the degree of impairment [16]. Subsequently, the rats were euthanized under deep anesthesia, and their brains were harvested and maintained in cold saline (4 °C). The olfactory bulb, cerebellum, and brainstem were removed, and the remaining brain tissue was briefly frozen at −20 °C for 10 min. Coronal brain slices were made (four cuts, yielding five slices in total), quickly immersed in 2% 2,3,5-triphenyltetrazolium chloride (TTC) solution, and incubated at 37 °C for 20 min in the dark. Viable brain tissues were stained a rose-red color, while ischemic tissues remained unstained. Each brain slice was digitally photographed, and the cerebral infarct area was quantitatively analyzed using the BI-2000 image analysis software,version 3.0 (Taimeng Medical Imaging, Chengdu, China). All the data are presented as the mean ± SD (*n* = 10 per group), and statistical comparisons were performed using a one-way ANOVA followed by Tukey’s post hoc test.

#### 4.5.5. The Water Content of Brain Tissue Was Measured

Following the ischemia–reperfusion process, the cerebellum, olfactory bulb, and lower brainstem were meticulously removed from the rats. The mass of the brain tissue left, referred to as the wet weight (W), was then accurately determined. This tissue was subsequently placed in an oven set at a constant temperature of 110 °C and dried for a duration of 48 h. Upon the completion of the drying period, the dry weight (D) of the tissue was recorded. The percentage of the water content in the brain tissue was calculated using the formula water content = [(W − D)/W] × 100%. All the data are presented as the mean ± SD (*n* = 10 per group), and statistical comparisons were performed using a one-way ANOVA followed by Tukey’s post hoc test.

#### 4.5.6. The Pathological Morphology of the Brain Tissue Was Observed

Following euthanasia, the brains of the rats were rapidly dissected and rinsed in cold saline at 4 °C. The olfactory bulb, cerebellum, and brainstem were removed. The brains were then fixed in 4% paraformaldehyde (Solarbio, Beijing, China) for 24–48 h, followed by dehydration in graded ethanol, clearing in xylene, embedding in paraffin, and coronal sectioning into 5 μm thick slices using a microtome (Leica RM2235, Leica Biosystems, Wetzlar, Germany). The sections were stained using standard hematoxylin and eosin (H&E) protocols. Hematoxylin stained cell nuclei dark blue, while eosin stained the cytoplasm and extracellular matrix pink. A microscopic examination was performed under a light microscope (Leica DM3000, Germany) to evaluate the degree of cellular damage, tissue disorganization, and neuronal loss in the ischemic regions. Representative images were captured, and pathological scoring or area quantification was performed using BI-2000 image analysis software,version 3.0 (Taimeng Medical Imaging, Chengdu, China). From each rat (*n* = 5 per group), three coronal sections were analyzed. For each section, three non-overlapping fields from the cortex and hippocampus were randomly selected and imaged at 200× magnification, yielding a total of nine images per animal for the statistical analysis. (*n* = 5; histological scores compared using a one-way ANOVA).

Histopathological evaluation focused on key morphological features indicative of ischemic injury, including neuronal cell loss, nuclear condensation and pyknosis, cytoplasmic eosinophilia, edema in the neuropil, vascular congestion or hemorrhage, and inflammatory cell infiltration. A semi-quantitative scoring system was used to assess tissue damage based on the extent and severity of histological alterations in the cortex and hippocampus. Each parameter was scored on a scale from 0 to 3 (0 = normal, 1 = mild, 2 = moderate, 3 = severe). All the sections were assessed blindly by two independent observers.

#### 4.5.7. The Activities of SOD, GSH-px, CAT, MDA, and LDH in the Brain Tissue Were Detected

Following the ischemia–reperfusion procedure in the rats, the olfactory bulb, cerebellum, and brainstem were removed. The remaining brain tissues were then finely chopped on ice. These tissues were mixed with physiological saline in a centrifuge tube at a specified ratio and homogenized using a tissue homogenizer until no visible solid particles remained. The mixture underwent centrifugation at 3000 rpm for 10 min, facilitating the separation of the supernatant, which was then carefully transferred into a fresh centrifuge tube. The concentrations of SOD, GSH-px, CAT, MDA, and LDH within the rat brain tissue were accurately measured using a multifunctional microplate reader [51]. All the data are presented as the mean ± SD (*n* = 10 per group), and statistical comparisons were performed using a one-way ANOVA followed by Tukey’s post hoc test.

#### 4.5.8. Detection of Nrf2, Keap1, HO-1, and NQO1 Protein Expression by Western Blot

The brain tissues were processed in a cell lysis buffer that included complete inhibitors for proteases and phosphatases. The nucleoproteins and total proteins were extracted, and their concentrations were determined using the BCA technique. Identical quantities of proteins from each group underwent separation via 10% SDS-PAGE and were subsequently transferred onto a PVDF membrane. The next complete steps follow the method in Section 4.4.12. All the data are presented as the mean ± SD (*n* = 3 per group), and statistical comparisons were performed using a one-way ANOVA followed by Tukey’s post hoc test.

#### 4.5.9. Statistical Analysis

The data from the experiments were processed and analyzed using the GraphPad Prism 9 software, which meant that all the results were presented as the mean ± SD. A one-way ANOVA was used for multiple group comparisons, followed by Tukey’s post hoc test to determine the statistical significance between all the group pairs (sham vs. model, SI vs. model, edaravone vs. model, and SI vs. edaravone). This approach ensured that every possible comparison between the study groups was systematically considered and appropriately tested. A *p*-value below 0.01 was interpreted as denoting a more substantial level of significance.

## 5. Conclusions

This study demonstrates that soy isoflavones exert a significant neuroprotective effect on PC12 cells damaged by Na_2_S_2_O_4_, enhancing neurological functions and reducing cerebral infarction in rats. These isoflavones notably enhance the activity of antioxidant enzymes, thereby mitigating oxidative stress resulting from ischemia–reperfusion. The underlying mechanisms of these protective effects are likely mediated through the activation of the Nrf2 signaling pathway.

## Figures and Tables

**Figure 1 pharmaceuticals-18-00548-f001:**
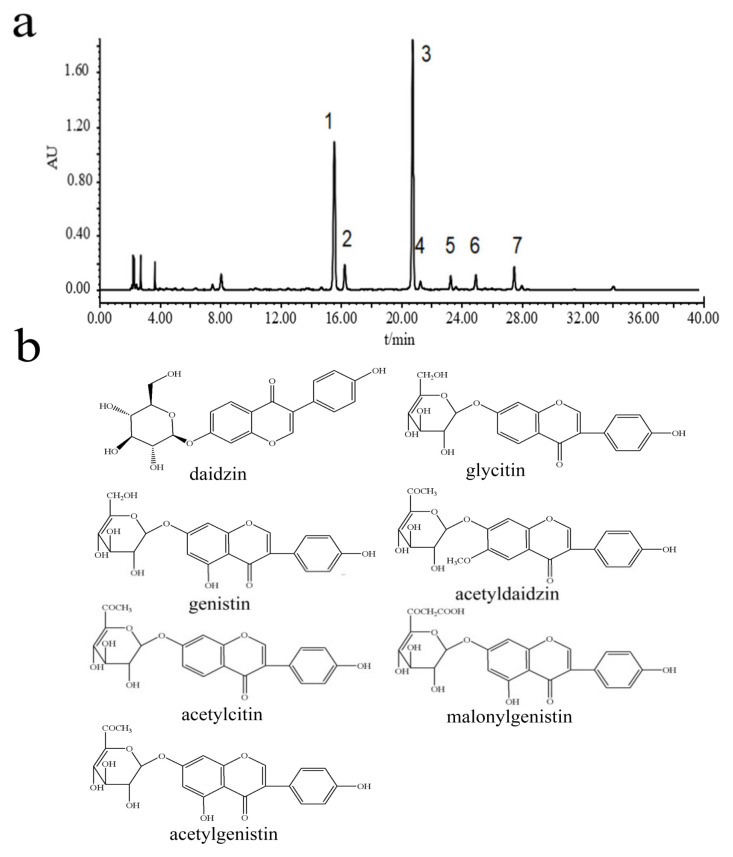
Analysis of crude soybean extract. (**a**) HPLC chromatogram. (**b**) Seven isoflavone structural formulas.

**Figure 2 pharmaceuticals-18-00548-f002:**
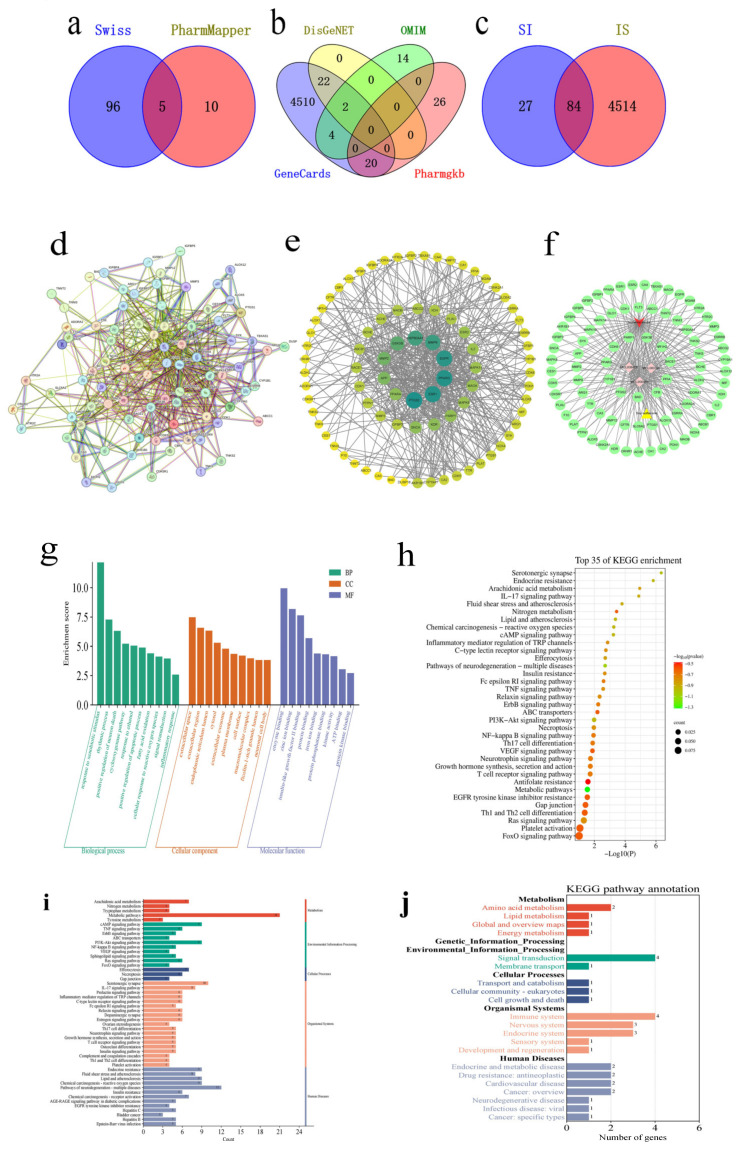
The exploration of the soy isoflavone (IS) network through a pharmacological analysis. (**a**) Drug Venn diagram. (**b**) Disease Venn diagram. (**c**) Drug–Disease Venn Diagram. (**d**) PPI network. (**e**) Beautified PPI network. (**f**) Drug–component–disease–target network. (**g**) GO enrichment analysis results. (**h**) KEGG pathway enrichment analysis findings. (**i**) A first-level taxonomic analysis of the KEGG enrichment pathway. (**j**) Secondary classification analysis of KEGG enrichment pathways.

**Figure 3 pharmaceuticals-18-00548-f003:**
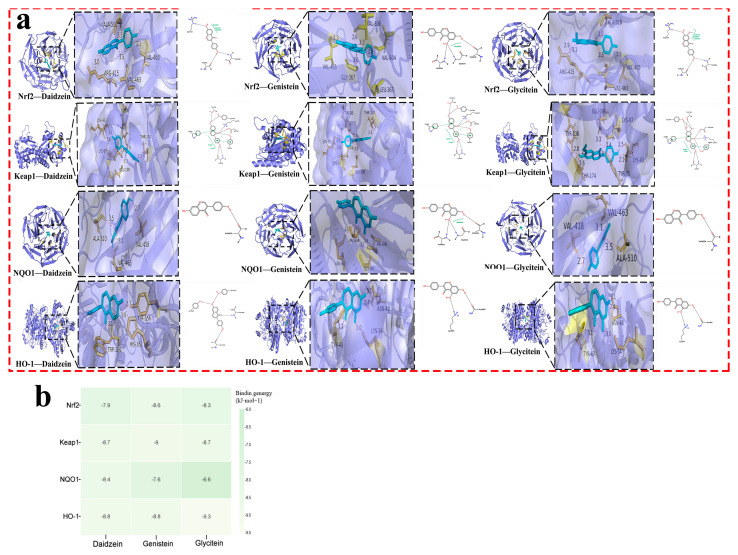
The molecular docking of the main active ingredients of soy isoflavones with key target proteins. (**a**) Visualization diagram of molecular docking. Ligands are shown in cyan, protein surfaces are shown in blue/purple, and yellow residues indicate key binding interactions. (**b**) Heat map. Darker green indicates stronger binding affinity (more negative binding energy, in kJ·mol^−1^).

**Figure 4 pharmaceuticals-18-00548-f004:**
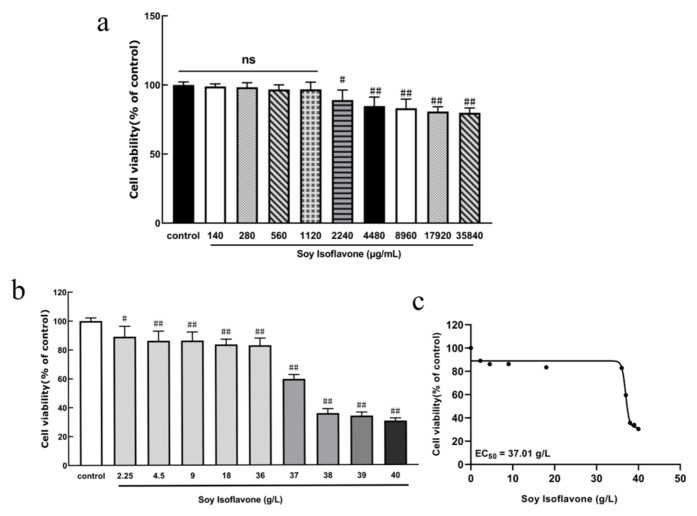
(**a**,**b**) Effects of different concentrations of soy isoflavones on cytotoxicity of PC12 vs. normal cell group. (Values are indicated as mean ± SD; statistical analysis was performed using one-way ANOVA followed by Tukey’s post hoc test; ns: no significant difference. *n* = 4, # *p* < 0.05, ## *p* < 0.01.) (**c**) Dose-response curve of soy isoflavone’s effect on PC12 cell viability. The normalized inhibition rate was calculated based on the control’s response. (EC_50_ value was estimated using Kaplan-Meier model. *n* = 4).

**Figure 5 pharmaceuticals-18-00548-f005:**
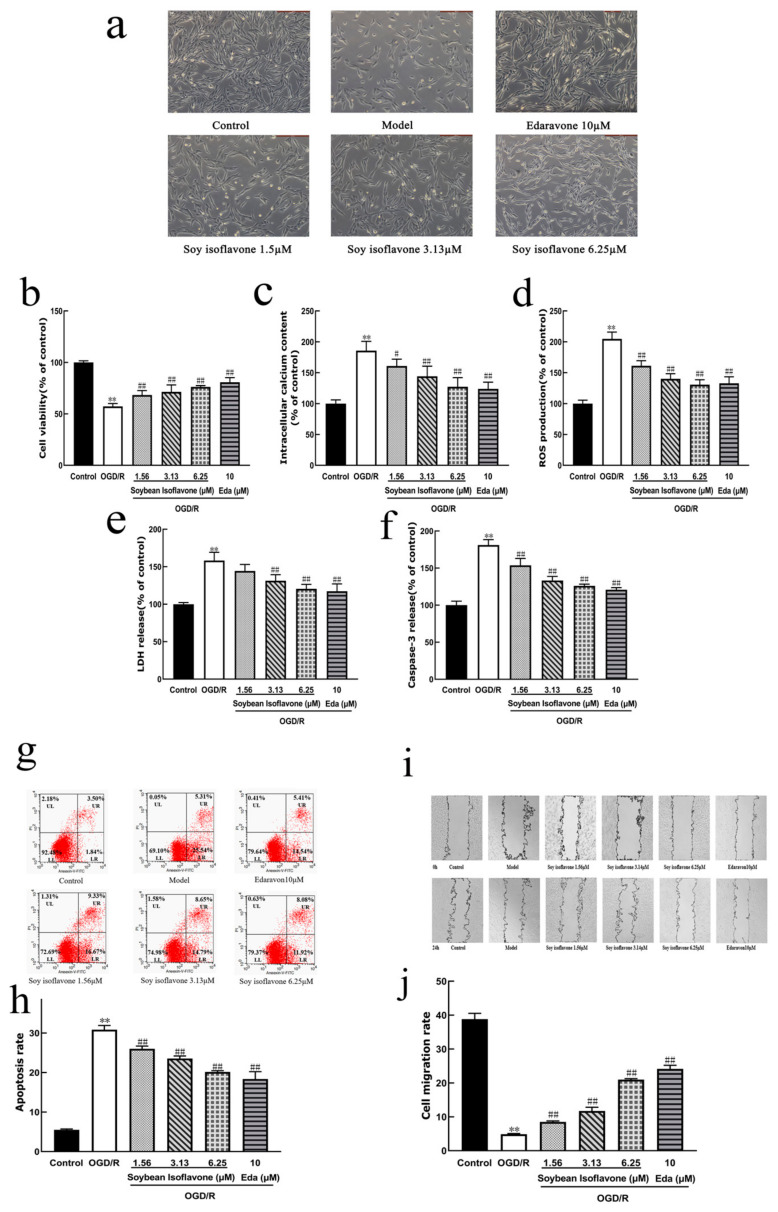
Effects of soybean isoflavones (SI) on PC12 cells under OGD/R conditions. (**a**) Representative morphological images of PC12 cells in each treatment group. Cells were exposed to different concentrations of soybean isoflavones (1.56, 3.13, and 6.25 μM) or edaravone (10 μM) following OGD/R induction. (**b**) Cell viability was assessed using the MTT assay. (**c**) Intracellular calcium levels were measured to evaluate calcium overload. (**d**) Reactive oxygen species (ROS) production was determined as an indicator of oxidative stress. (**e**) Lactate dehydrogenase (LDH) release was measured as a marker of membrane damage. (**f**) Caspase-3 release was detected to assess apoptosis activation. (**g**) Flow cytometric analysis of apoptosis using Annexin V-FITC/PI staining. (**h**) Quantification of apoptosis rate based on flow cytometry results. (**i**) Representative images of the wound healing assay to evaluate PC12 cell migration. (**j**) Quantification of cell migration rate. All values are expressed as mean ± SD. Statistical significance was determined by one-way ANOVA followed by Tukey’s post hoc test. (Values are indicated as mean ± SD; statistical analysis was performed using one-way ANOVA followed by Tukey’s test. *n* = 6 for (**b**)–(**f**), *n* = 3 for (**h**)–(**j**). ** *p* < 0.01, compared to control, # *p* < 0.05, ## *p* < 0.01 versus model cohort).

**Figure 6 pharmaceuticals-18-00548-f006:**
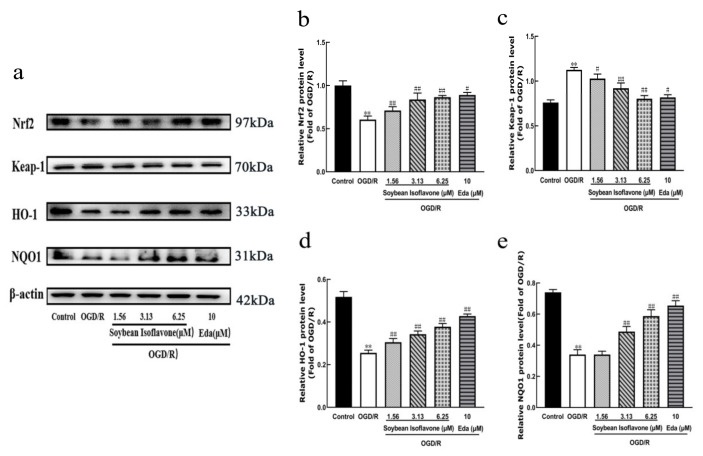
Examination of soy isoflavone (SI)’s influence on oxygen–glucose deprivation/reoxygenation (OGD/R) protein expression in PC12 cells. (**a**) Western blotting analysis. (**b**) Comparative expression of nuclear factor erythroid 2-related factor 2 (Nrf2) protein. (**c**) Kelch-like, ECH-associated protein 1 (Keap1) protein expression levels. (**d**) Heme oxygenase-1 (HO-1) protein expression. (**e**) NAD(P)H quinone dehydrogenase 1 (NQO1) protein expression levels. (Values are indicated as mean ± SD; statistical analysis was performed using one-way ANOVA followed by Tukey’s post hoc test. *n* = 3, ** *p* < 0.01, compared to control, # *p* < 0.05, ## *p* < 0.01 versus model cohort).

**Figure 7 pharmaceuticals-18-00548-f007:**
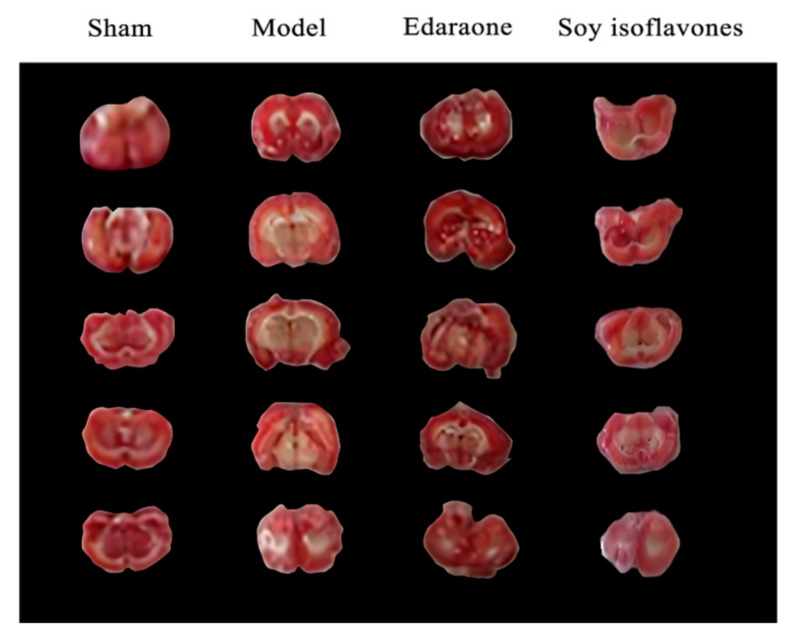
The impact of Tectoridin on neurological deficits.

**Figure 8 pharmaceuticals-18-00548-f008:**
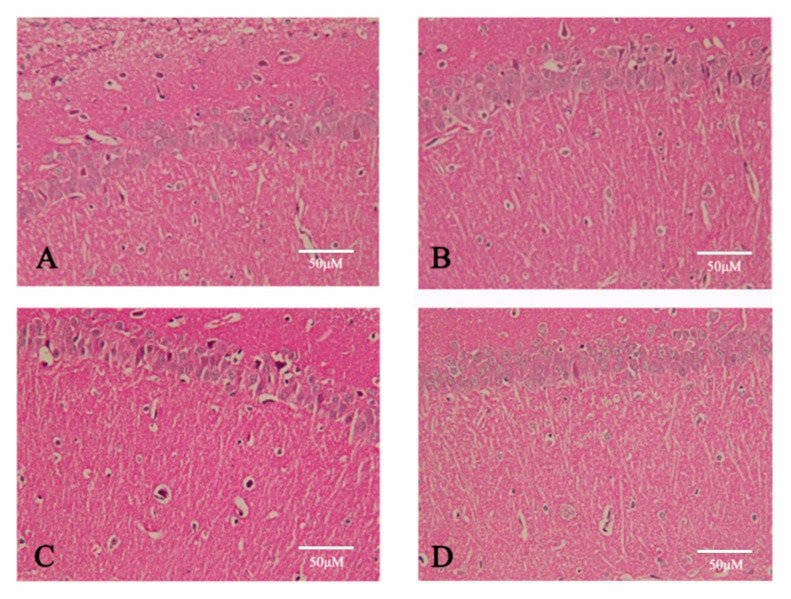
Pathological alterations in brain tissue across different experimental groups. (**A**) Sham-operated group. (**B**) Model. (**C**) Treated with soybean isoflavones (100 mg/kg). (**D**) Treated with edaravone (10 mg/kg).

**Figure 9 pharmaceuticals-18-00548-f009:**
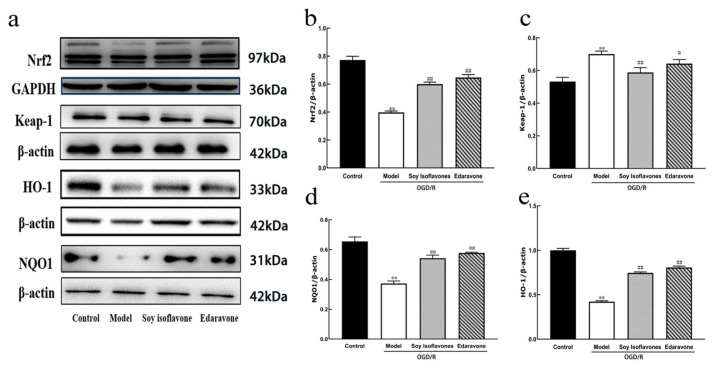
Soy isoflavones (SI) reduce oxidative stress injury by increasing nuclear factor erythroid 2-related factor 2 (Nrf2) signaling pathway. (**a**) Western blot representation of nuclear Nrf2 levels and expression of Kelch-like, ECH-associated protein 1 (Keap1), heme oxygenase-1 (HO-1), and NAD(P)H quinone dehydrogenase 1 (NQO1). (**b**) Comparative expression of Nrf2 protein. (**c**) Keap1 protein expression levels. (**d**) NQO1 protein expression. (**e**) HO-1 protein expression levels. (Values are indicated as mean ± SD; statistical analysis was performed using one-way ANOVA followed by Tukey’s post hoc test. *n* = 4, ** *p* < 0.01, compared to control, # *p* < 0.05, ## *p* < 0.01 versus model cohort).

**Figure 10 pharmaceuticals-18-00548-f010:**
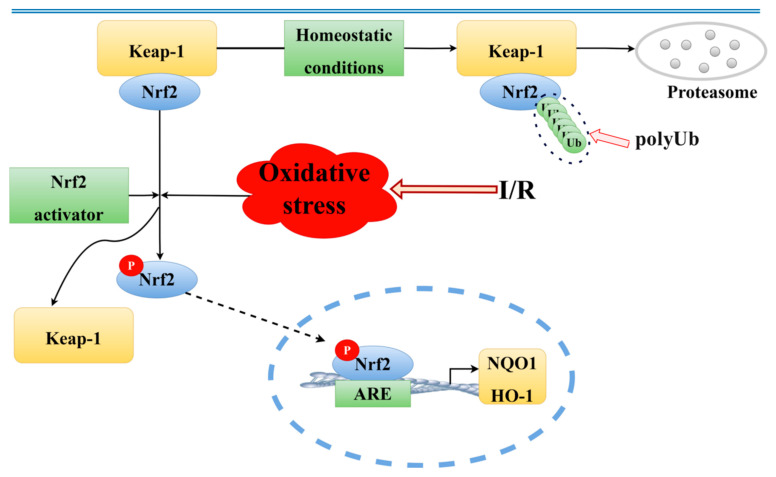
The molecular mechanism of the Keap1/NQO1/Nrf2/HO-1 pathway. Solid black arrows indicate biological processes; dashed black arrows represent translocation.

**Table 1 pharmaceuticals-18-00548-t001:** LC-MS analysis data of soybean’s active ingredients Comprehensive LC-MS data for identifying active compounds in soybeans.

NO.	Retention Time (min)	MS (m/z)	MS2 (m/z)	Compound
1	15.4	417, [M + H]^+^	255, 137	Daidzin
2	16.2	447, [M + H]^+^	285, 270	Glycitin
3	20.7	433, [M + H]^+^	271, 153	Genistin
4	21.4	459, [M + H]^+^	255	Acetyldaidzin
5	23.3	489, [M + H]^+^	285	Acetylcitin
6	25.2	519, [M + H]^+^	271, 215, 197, 153	Malonylgenistin
7	27.5	475, [M + H]^+^	271	Acetylgenistin

**Table 2 pharmaceuticals-18-00548-t002:** Existing forms and structural formulas of 12 soy isoflavones.

Type	Types of Isoflavones	R1	R2	R3
Free type	Daidzein	H	H	
	Glycitein	H	OCH_3_	
	Genistein	OH	H	
Glucoside type	Daidzin	H	H	H
	Genistein	H	OCH_3_	H
	Glycitin	OH	H	H
Acetyl glucoside type	Acetyldaidzin	H	H	COCH_3_
	Acetylglycitin	H	OCH_3_	COCH_3_
	Acetylgenistin	OH	H	COCH_3_
Malonylglucoside type	Malonyldaidzin	H	H	COCH_2_COOH
	Malonylglycitin	H	OCH_3_	COCH_2_COOH
	Malonylgenistin	OH	H	COCH_2_COOH

**Table 3 pharmaceuticals-18-00548-t003:** Comparative assessment of neurological function, brain water content, and cerebral infarct volume in rat groups, indicating statistical significance. (Values are indicated as mean ± SD; statistical analysis was performed using one-way ANOVA followed by Tukey’s post hoc test. *n* = 10, ** *p* < 0.01, compared to control, # *p* < 0.05, ## *p* < 0.01 versus model cohort).

Group	Neurological Function Score	Brain Tissue Water Content	Cerebral Infarct Volume
Sham operation group	0 ± 0	78.5 ± 3.6	0 ± 0
Model group	8.3 ± 1.09 **	85.4 ± 2.6 **	27.6 ± 1.6 **
Edaravone	5.4 ± 2.11 ##	77.5 ± 2.4 ##	23.9 ± 1.4 ##
Soybean isoflavone	6.8 ± 1.68 ##	76.5 ± 3.2 ##	22.5 ± 2.1 #

**Table 4 pharmaceuticals-18-00548-t004:** Evaluation of antioxidant activities (SOD, GSH-Px, CAT) and levels of oxidative stress markers (MDA, LDH) in the brain tissue of rat groups, with significance denoted. (Values are indicated as mean ± SD; statistical analysis was performed using one-way ANOVA followed by Tukey’s post hoc test. *n* = 4, ** *p* < 0.01, compared to control, ## *p* < 0.01 versus model cohort).

Group	SOD (U/mg prot)	GSH-px (U/mg prot)	CAT (U/mg prot)	MDA (nmol/mg)	LDH (U/L)
Sham operation group	18.5 ± 1.6	26.7 ± 2.6	5.6 ± 1.2	3.8 ± 0.6	180.8 ± 15.3
Model group	9.4 ± 1.6 **	9.8 ± 1.6 **	2.3 ± 1.1 **	6.1 ± 1.1 **	319.3 ± 25.1 **
Edaravone	12.5 ± 1.4 ##	13.9 ± 1.4 ##	3.3 ± 1.3 ##	4.4 ± 1.2 ##	216.4 ± 22.21 ##
Soybean isoflavone	10.5 ± 1.2 ##	12.5 ± 2.1 ##	2.9 ± 1.4 ##	4.2 ± 1.6 ##	193.3 ± 11.7 ##

## Data Availability

The data that support the findings of this study are available within the article.

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
