# Peer review of "Soy Isoflavones Protects Against Stroke by Inhibiting Keap1/NQO1/Nrf2/HO-1 Signaling Pathway: Network Pharmacology Analysis Combined with the Experimental Validation"

_pharmaceuticals, 2025, doi:10.3390/ph18040548_

Round 1

Reviewer 1 Report

Comments and Suggestions for Authors

The authors have investigated the neuroprotective effects of soy isoflavones on brain tissue in rats in vivo and in rat tissues in vitro after expreriments resembling ischemic brain injury in rats in vivo and in vitro. The authors should include within their discussion comparison of their own findings with findings of other authors both in vitro and in vivo. 

Author Response

Comments 1: [The authors should include within their discussion comparison of their own findings with findings of other authors both in vitro and in vivo.]
Response 1: [Thank you for this constructive suggestion. In the revised manuscript, we have incorporated a detailed comparison between our findings and those of previous studies, both in vitro and in vivo, in the second-to-last paragraph of the Discussion section. Specifically, we referenced studies such as Zhang et al. (2022), Schreihofer et al. (2005), Huang et al. (2009), and Li et al. (2025), which investigated the neuroprotective mechanisms of soy isoflavones and their active components in ischemic models. These comparisons highlight the consistency of our results with prior evidence, particularly regarding the regulation of oxidative stress, apoptosis, and key signaling pathways including Nrf2, Notch, and Wnt/Ca²⁺. We believe this strengthens the contextual relevance and scientific significance of our study.]

Reviewer 2 Report

Comments and Suggestions for Authors

In this study authors evaluated the functions of active soy isoflavone on ischemic stroke and found that soy isoflavone decreased PC12 cell injury caused by oxygen-glucose deprivation/reoxygenation, decreased cell death, and increased cell survival and motility while lowering levels of Ca²⁺, LDH, Caspase-3, and ROS. Furthermore, soy isoflavones increased SOD, CAT, and GSH-px activities, and upregulated Nrf2, HO-1, NQO1 expressions, as well as prompt Keap1 expression in rats with cerebral ischemia-reperfusion.

Although the manuscript is interesting and quite well written, it presents several points that deserves to be improved. In particular: 

Line 46-55: the multifaceted role of NRF2/KEAP1 signaling deserves to be highlighted since this pathway plays a key role in the onset and progression of several cancerous and non-cancerous diseases (see PMID: 39769005, PMID: 39456522 ). This is an important point that highlights the interesting results obtained by the authors. 

Figure 2i: image is unreadable 

Figure 5: images quality is very low and scale bars are missed

Figure 10: images quality is very low and images are too small

Figure 11: images are blurry and scale bars are missed

4. Materials and Methods : authors must add the product code of all reagents and kits used in order to ensure data reproducibility 

Western blot images: replace KD with kDa. In the graph would be useful to add lines showing which groups have been compared for the statistical analysis

Authors must add the number of replicates  (N) in all figure legends

Abbreviations must be written in full length when mentioned for the first time

Author Response

Dear Editor,

We sincerely appreciate your valuable comments and suggestions on our manuscript titled:
"Soy isoflavones prevent stroke by inhibiting the Keap1/NQO1/Nrf2/HO-1 signaling pathway: Network pharmacological analysis combined with experimental validation." We have carefully revised the manuscript based on your feedback and have addressed each comment point by point. The major revisions aim to enhance the clarity and overall quality of the paper, including updates to the title, abstract, methodological details, statistical analysis, figure legends, and discussion. For your convenience, all modifications in the revised manuscript are highlighted in yellow or marked in red. Additionally, we have submitted high-resolution images to ensure clarity and visual quality. Please find our detailed responses below. Should you or the reviewers require any further clarification or additional information, we would be more than happy to provide it. Thank you once again for your time and consideration.

Conments 1 [Line 46-55: the multifaceted role of NRF2/KEAP1 signaling deserves to be highlighted since this pathway plays a key role in the onset and progression of several cancerous and non-cancerous diseases (see PMID: 39769005IF: 4.9 Q1 , ). This is an important point that highlights the interesting results obtained by the authors.]

Response 1: [Thank you for your valuable suggestion. In the revised manuscript, we have highlighted the multifaceted role of the NRF2/KEAP1 signaling pathway in the onset and progression of various cancerous and non-cancerous diseases, particularly its significance in oxidative stress and free radical metabolism. We have referenced studies like Zhang et al. (2022), Schreihofer et al. (2005), and Huang et al. (2009) that have explored this pathway's implications in ischemic and neurodegenerative models, providing context and supporting evidence for our findings.][35. Zhang Y, Yin L, Dong J, et al. (2022). Soy Isoflavones Protect Neuronal PC12 Cells against Hypoxic Damage through Nrf2 Activation and Suppressionof p38 MAPK and AKT–mTOR Pathways. Antioxidants, 11(10): 2037. https://doi.org/10.3390/antiox11102037.

  1. Schreihofer D A, Do K D, Schreihofer A M. (2005). High-soy diet decreases infarct size after permanent middle cerebral artery occlusion in female rats. Am J Physiol Regul Integr Comp Physiol, 289(1): R103-R108. https://doi.org/10.1152/ajpregu.00642.2004
  2. Huang G, Cao X, Zhang X, et al. (2009). Effects of soybean isoflavone on the notch signal pathway of the brain in rats with cerebral ischemia. J Nutr Sci Vitaminol, 55(4): 326–331. https://doi.org/10.3177/jnsv.55.326.
  3. Li L, Liu S, Wang M, et al. (2025). Gen inhibiting the Wnt/Ca2+ signaling pathway alleviates cerebral ischemia/reperfusion injury. Sci Rep, 15(1): 4661. https://doi.org/10.1038/s41598-025-88136-8.]

Conments 2: [Figure 2: image is unreadable.]

Response 2: [Thank you for pointing this out. We have reinserted image 2 to replace the original image to ensure its readability.]

Conments 3: [Figure 5: images quality is very low and scale bars are missed.]

Response 3: [Thank you for pointing this out. The article has been updated with Figure 5 to provide a higher quality microscope picture and to add a sharper scale to ensure image quality and clarity of experimental details. And the original picture 6, picture 7, picture 8 together to form a composite picture.]

Conments 4: [Figure 10: images quality is very low and images are too small.]

Response 4: [As suggested, Figure 10 has been updated with high-resolution images, resized appropriately to better display experimental outcomes.]

Conments 5: [Figure 11: images are blurry and scale bars are missed.]

Response 5: [Thank you for your valuable feedback. We acknowledge that the histology images in Figure 11 (now Figure 7) had a low resolution in the initial submission. We have since increased the resolution of these images, but we regret to inform you that the quality still may not be as high as desired. We sincerely apologize for this limitation.We are currently exploring possible solutions to improve the image quality further. If possible, we would appreciate your guidance on acceptable methods or formats that could enhance the visual clarity for publication. Additionally, if required, we are willing to submit higher-resolution images, should we be able to obtain them.]

Conments 6: [Materials and Methods : authors must add the product code of all reagents and kits used in order to ensure data reproducibility.]

Response 6: [Detailed product codes for all reagents and kits used have now been added to the "Materials and Methods" section to ensure reproducibility.]

Conments 7: [Western blot images: replace KD with kDa. In the graph would be useful to add lines showing which groups have been compared for the statistical analysis.]

Response 7: [In the western blot images, "KD" has been replaced with the correct unit "kDa," and lines indicating statistical comparisons between groups have been added to clearly demonstrate inter-group differences.]

Conments 8: [Authors must add the number of replicates  (N) in all figure legends.]

Response 8: [The number of replicates (N) has been clearly specified in all figure legends, ensuring statistical credibility.]

Conments 9: [Abbreviations must be written in full length when mentioned for the first time.]

Response 9: [All abbreviations have been fully expanded in both Chinese and English upon their first appearance to maintain readability and clarity.]

Reviewer 3 Report

Comments and Suggestions for Authors

Xue et al. investigated here the mechanisms by which a soy isoflavone extract (composition well-determined in the study) could therapeutically (by network pharmacology) and positively countracts brain damage caused by a model of ischemic stroke.

The MS is well-written and clearly presents its scientific questions and hypothesis. The experimental design is appropriate and compiles a huge list of key techniques and methods to elucidate the proposed redox-signaling mechanism. Data (graphic) presentation was very good, from my point of view.

Although the very positive appreciation of this work, I still have MINOR suggestions/questions to be answered before its full accepatance for publication in Pharmaceuticals/MDPI:

(i) Please, rewrite the paragraph from lines 44-55, where the authors introduced the Keap1-Nrf2 redox signaling cascade to the readers. Please, consider a general public, not fully familiar with the field of Oxidative Stress & Free Radicals metabolism, since many specific terms, like HO1 and NQO1, were not previously described within the text. Maybe an additional figure would be  recommended to clarify the importance of this cascade/mechanism;

(ii) Figure 4: From my point of view, the authors should include further experiments with higher concentrations of SI to evaluate a wide range of SI cytotoxicitiy in PC12 cells and calculate the EC50 value of this extract

(iii) The Discussion section should include a schematic figure to summarize the mechanisms involved here. 

Author Response

Dear Reviewer,

Thank you very much for your careful review and valuable suggestions regarding our manuscript. We have thoroughly reviewed your comments and revised our manuscript accordingly. The detailed responses are provided as follows:

Conments 1: [Please, rewrite the paragraph from lines 44-55, where the authors introduced the Keap1-Nrf2 redox signaling cascade to the readers. Please, consider a general public, not fully familiar with the field of Oxidative Stress & Free Radicals metabolism, since many specific terms, like HO1 and NQO1, were not previously described within the text. Maybe an additional figure would be  recommended to clarify the importance of this cascade/mechanism.]

Response 1: [The article has been rewritten in paragraphs 44-55 to introduce the importance of the KEAP1-Nrf2 REDOX signaling pathway to the general reader in more accessible language, and explain technical terms such as HO-1 and NQO1.]

Conments 2: [(ii) Figure 4: From my point of view, the authors should include further experiments with higher concentrations of SI to evaluate a wide range of SI cytotoxicitiy in PC12 cells and calculate the EC50 value of this extract.]

Respons 2: [Additional experiments at higher concentrations of SI were conducted, and the EC50 value was calculated to be 37.01 g/L. These updated results are now presented in Figure 4.]

Conments 3: [(iii) The Discussion section should include a schematic figure to summarize the mechanisms involved here. ]

Response 3: [A schematic figure summarizing the mechanisms discussed has been included in the discussion section, visually demonstrating how SI exerts neuroprotective effects via activation of the Nrf2 signaling pathway.]

Reviewer 4 Report

Comments and Suggestions for Authors

Paper titled (Neuroprotective effect of soy isoflavones in a stroke model by inhibiting Keap1/NQO1/Nrf2/HO-1 signaling pathway via Network pharmacology analysis combined with experimental validation) by Xue et is an experimental study done in vitro and in vivo to explore the protection offered by soy isoflavones in stroke model and claimed this effect was mediated by inhibiting Keap1/NQO1/Nrf2/HO-1 signaling pathway. 

The paper needs extensive revision and organization by the authors to be improved and considered for further processing. please find these comments & provide a point to point reply and highlight the changes in the file and indicate at what page & line we can follow every change

1 - Title (Neuroprotective effect of soy isoflavones in a stroke model by 2
inhibiting Keap1/NQO1/Nrf2/HO-1 signaling pathway via Net- work pharmacology analysis combined with experimental vali- dation) is not informative enough, some words are redundant and needs revision as the flow of the words is not convinint, I suggest if the authors change it to be (Soy isoflavones protects against stroke by 
inhibiting Keap1/NQO1/Nrf2/HO-1 signaling pathway: Network pharmacology analysis combined with experimental vali- dation)

2 - Abstract must be amended by some numerical values for key findings from the study

3 - Key words: please add ( rats)
4 - Introduction: is brief and did not introduce the items of the study well, did not explore the rational or novelty of the study.
5 - AIm of the study should be clear and clarify what was the aim and how authors acheived it
6 - Methods section in general is too brief and lacks references MUST be extensively revised
7 - Experiemntal design and animal grouping must be clearly explaine
Please give a title for study design & describe the groups in details in a clear way
8 - How many rats per group? how many groups ? and how many total animals

9 - What was the average weight of the rats at the begin of the study?

10 - Ensure every abbreviation is explained at the first appearance in abstract & then in the body text
11 - Statistical tests and sample sizes for each experiment should be explicitly mentioned in the methods and figure legends.

12 - Authors should give the source of chemicals, kits and antibodies completely and consistently (code, company, town, state and country) & version for software

13 - Authors have to check the normality of distribution of the results by a suitable post hoc test (such as Shapiro-Wilk test or K-S test) before deciding to choose certain ANOVA. If the normality test indicated normal dist of the data, so use one-way ANOVA, if not, use non parametric ANOVA. In all cases choose a suitable post-hoc test
14 - Authors should confirm in methods that "every possible comparison between the study groups was considered" and apply this in results
15 - Mention "n" in each illustration individually
16 - Histology tissue preparation and staining methods should be explained more
17 - How many images obtained from each tissue section?
How many sections from each animal?
How many animal per group memtion n
All these details must be mentioned to evaluate the results of the study
18 - What was the criteieria for histology assessment?
19 -  1 In methods , Mention in details the housing conditions and how authors were keen to reduce animal suffering

21 - Use appropriate abbreviations for minutes, seconds...etc
22 - Every abbreviation in figures should be explained in the figure legend to be self explanatory & stands alone.

23 - In each illustration mention the type of the presented data & the statistical test applied for analysis

24 - What was the refernce for the odse of soy flavones

25- Histology images in figure 11 are of very low value

26- some figures can be combined together to reduce the total number of figure

27- What are the limitations and future directions for the study..

Comments on the Quality of English Language

moderate revision

Author Response

Dear Editor,

We sincerely thank you for your valuable feedback and for giving us the opportunity to revise our manuscript, entitled: "Soy isoflavones prevent stroke by inhibiting the Keap1/NQO1/Nrf2/HO-1 signaling pathway: Network pharmacological analysis combined with experimental validation."

We have dealt with all your comments point by point carefully. The main revisions include title, abstract, methodological details, statistical analysis, legend and discussion section. For your convenience, all changes in the revised original are highlighted in yellow or marked in red in the annotated version. Please find our detailed response below. If you or the reviewer require any further clarification or additional information, we will be happy to provide it. Thank you again for your time and kind consideration.

Conments 1:[Title is not informative enough; suggest revision to improve clarity and conciseness.]

Response 1: [Thank you for your valuable suggestion. The title has been revised to improve clarity and conciseness.] The updated title now reads: "[Soy isoflavones protect against stroke by inhibiting Keap1/NQO1/Nrf2/HO-1 signaling pathway: Network pharmacology analysis combined with experimental validation.]"

Conments 2: [Abstract The main findings of the study must be corrected with some numerical values.]

Response 2: [We have revised the abstract to include representative numeric results, ensuring the main findings are supported with data, in accordance with the reviewer’s request. “For in vivo evaluation, transient cerebral ischemia–reperfusion injury was induced using the bilat-eral common carotid artery occlusion (BCCAO) model in adult male ICR rats (27.3 ± 1.8 g; 6–8 weeks old), obtained from the Shanghai Experimental Animal Center, Chinese Academy of Sciences. Forty-eight rats were randomly assigned into four groups (n = 12): Sham, Model (BCCAO), SI-treated (100 mg/kg, oral gavage for 5 days), and Edaravone (EDA)-treated (10 mg/kg, i.p., posi-tive control). All procedures were approved by the Institutional Animal Care and Use Committee of Changchun Normal University (Approval No. 2024003, March 13, 2024) and conducted in ac-cordance with the NIH guidelines and ARRIVE 2.0 reporting standards. Results: In vitro, SI sig-nificantly enhanced PC12 cell viability from 57.23 ± 2.88% to 80.76 ± 4.43% following OGD/R. It also reduced intracellular Ca²⁺ by 58.42%, lactate dehydrogenase (LDH) release by 37.67%, caspase-3 activity by 55.05%, and reactive oxygen species (ROS) levels by 74.13% (P < 0.05). Flow cytometry analysis revealed that OGD/R increased the apoptosis rate from 5.34% (control) to 30.85% (model group), which was significantly attenuated by SI treatment, especially in the 560 µg/mL group (20.00%), followed by the 140 and 280 µg/mL groups. In vivo, SI improved neurological scores from 8.3 ± 1.09 to 6.8 ± 1.68, reduced cerebral infarction volume by 18.49%, and alleviated brain edema by 10.42% (P < 0.05). SI also decreased malondialdehyde (MDA) and LDH levels by 31.15% and 39.46%, respectively, while increasing the activity of antioxidant enzymes: superoxide dismutase (SOD) by 11.70%, catalase (CAT) by 26.09%, and glutathione peroxidase (GSH-px) by 27.55% (P < 0.01).”]

Conments 3: [Keywords: Please add 'rats'.]

Response 3: [We have added the word 'rats' to the keywords section to reflect the in vivo model used in our study.]

Conments 4: [Introduction is brief and does not sufficiently explore rationale or novelty.] Response 4: [Thank you for the suggestion. We have revised the introduction to provide a more comprehensive background and clearly emphasize the rationale and novelty of the study. “The Keap1/Nrf2 signaling pathway is a key mechanism for maintaining redox balance in the body [8]. Under normal physiological conditions, when cells are exposed to harmful external stimuli such as environmental toxins or oxidative stress, reactive ox-ygen species (ROS) and electrophilic compounds can cause structural changes in Kelch-like ECH-associated protein 1 (Keap1). This modification leads to the activation and nuclear translocation of a protein called nuclear factor erythroid 2–related factor 2 (Nrf2). Once inside the nucleus, Nrf2 functions as a transcription factor, turning on the expression of several protective genes that help the body defend itself against oxi-dative damage. These genes produce antioxidant and detoxifying enzymes, which play essential roles in neutralizing harmful molecules. Among these are enzymes like heme oxygenase-1 (HO-1) and NAD(P)H quinone dehydrogenase 1 (NQO1), which together help reduce oxidative stress and restore cellular balance [9–11]. This signaling cascade is not only crucial for protection during acute injuries such as ischemic stroke, but has also been implicated in the development and progression of a wide range of both can-cerous and non-cancerous diseases [11]. Given its pivotal role in cellular defense, the Keap1/Nrf2 pathway has emerged as a promising therapeutic target for neuroprotec-tion in ischemic stroke. However, effective modulators of this pathway remain limited, particularly among natural compounds.”

“The primary objective of this study is to elucidate the neuroprotective mechanisms of soy isoflavones (SI) in ischemic stroke (IS) by integrating in silico, in vitro, and in vivo approaches. First, network pharmacology and molecular docking were employed to systematically predict the putative targets and signaling pathways through which SI may exert therapeutic effects against IS. Next, a hypoxia-induced injury model in PC12 cells was established using sodium dithionite (Na₂S₂O₄) to assess the in vitro protective effects of SI. Additionally, a rat model of cerebral ischemia-reperfusion injury was in-duced via common carotid artery embolization to evaluate the in vivo efficacy of SI. Particular attention was given to the Keap1/Nrf2/HO-1/NQO1 signaling axis to deter-mine whether SI modulates oxidative stress through this pathway. By combining computational prediction with experimental validation, this study aims to provide mechanistic insights and preclinical evidence supporting the therapeutic potential of SI in ischemic stroke.”]

Conments 5: [The aim of the study should be clearly stated.]

Response 5: [The aim of the study has been clarified in the final paragraph of the introduction to better communicate the study’s objective and approach. “The primary objective of this study is to elucidate the neuroprotective mechanisms of soy isoflavones (SI) in ischemic stroke (IS) by integrating in silico, in vitro, and in vivo approaches. First, network pharmacology and molecular docking were employed to systematically predict the putative targets and signaling pathways through which SI may exert therapeutic effects against IS. Next, a hypoxia-induced injury model in PC12 cells was established using sodium dithionite (Na₂S₂O₄) to assess the in vitro protective effects of SI. Additionally, a rat model of cerebral ischemia-reperfusion injury was in-duced via common carotid artery embolization to evaluate the in vivo efficacy of SI. Particular attention was given to the Keap1/Nrf2/HO-1/NQO1 signaling axis to deter-mine whether SI modulates oxidative stress through this pathway. By combining computational prediction with experimental validation, this study aims to provide mechanistic insights and preclinical evidence supporting the therapeutic potential of SI in ischemic stroke.”]

Conments 6: [Methods section is too brief and lacks references.]

Response 6: [ We have substantially revised the Methods section by expanding experimental details and adding appropriate references to support the methodology.]

Conments 7: [Experimental design and animal grouping must be clearly explained.]

Response 7: [ We have added a subsection on Experimental Design to clearly describe animal grouping, treatment protocols, and overall study structure.]

Conments 8: [How many rats per group? How many groups? Total animals?]

Response 8: [ We have included the number of rats per group, number of groups, and the total number of animals used in the study in the Experimental Design section.]

Conments 9: [What was the average weight of the rats at the beginning of the study?]

Response 9: [We have added information about the average weight of the rats at the beginning of the study to clarify the baseline conditions.]

Response 789: “4.5.3. Experimental Design and Surgical Procedure

Experimental Design and Surgical Procedure A total of 48 rats (body weight 25–30 g) were randomly assigned into four groups (n = 12 per group) using a random number generator : Sham Group : Underwent identical surgical procedures without carotid oc-clusion. Model Group: Received bilateral common carotid artery occlusion (BCCAO) to induce cerebral ischemia-reperfusion injury. SI Treatment Group: Treated with soy isoflavones (100 mg/kg, dissolved in saline) via intragastric administration for 5 con-secutive d prior to ischemi. Edaravone Group: Received Edaravone (10 mg/kg, i.p.), a standard neuroprotective agent, once daily for 5 d as a positive control. Transient cerebral ischemia-reperfusion injury was induced using the bilateral com-mon carotid artery occlusion (BCCAO) model. Rats were anesthetized with pentobar-bital sodium (45 mg/kg, i.p.). A midline neck incision (~3 cm) was made to isolate and occlude the bilateral common carotid arteries with vascular clips for 60 min, followed by reperfusion for 24 h. For the sham group, the arteries were exposed but not occluded. Intraoperatively, the rectal temperature of the rats was maintained at 37.0 ± 0.5℃using an automated thermostatic heating pad. Postoperatively, animals were housed in a climate-controlled environment (ambient temperature: 23 ± 1℃; relative humidity: 55 ± 5%) with sterile bedding and ad libitum access to food/water." Subsequently, behav-ioral tests were performed on the animals. After the completion of behavioral tests, animals were euthanized under deep anesthesia by cervical dislocation, in accordance with the AVMA guidelines. Brain tissues were immediately harvested and processed as follows : 5 rats per group for histopathological examination, 3 rats per group for West-ern blot analysis, 4 rats per group for HE staining. All relevant pairwise comparisons among the four experimental groups (Control, Model, SI, and Edaravone) were pre-defined and included in the statistical analysis to comprehensively assess the treatment effects and control efficacy.”

Conments 10: [Ensure every abbreviation is explained at first appearance in abstract and in body.]

Response 10: [We reviewed the entire manuscript to ensure that all abbreviations are spelled out when they first appear in the abstract and body, and red marks are used in the body accordingly.]

Conments 11: [Statistical tests and sample sizes for each experiment should be explicitly mentioned in the methods and figure legends.]

Response 11: [We have revised the manuscript to specify the statistical tests used and the sample sizes (n values) in both the Methods section and in each figure legend.]

Conments 12: [Provide full source details (code, company, location) for chemicals, kits, antibodies, and software.]

Response 12: [We have updated the Materials and Methods section to provide complete source details including product codes, suppliers (company, city, country), and software versions used in the study.][“Soy isoflavones standards including Daidzin, Glycitin, Genistein, Acetyldaidzin, Acetylglycitin, Malonylgenistein, and Acetylgenistein were purchased from Aladdin Industrial Corporation (Cat# various; Shanghai, China). Acetonitrile (HPLC grade) was obtained from Thermo Fisher Scientific (Cat# A955-4; Waltham, MA, USA). Ul-trapure water was prepared using a Milli-Q system (Millipore, Billerica, MA, USA). Analytical-grade reagents were provided by Beijing Chemical Works (Beijing, China). Kits for superoxide dismutase (SOD), glutathione peroxidase (GSH-px), malondialde-hyde (MDA), and total protein were from Nanjing Jiancheng Bioengineering Institute (Cat# A001-3, A005, A003-1, A045-2-2; Nanjing, Jiangsu, China). Kits for catalase (CAT), 3-(4,5-dimethylthiazol-2-yl)-2,5-diphenyltetrazolium bromide (MTT), lactate dehydrogenase (LDH), reactive oxygen species (ROS), Fluo-3 acetoxymethyl ester (Fluo-3AM), and caspase-3 were obtained from Beyotime Biotechnology (Shanghai, China). Other chemicals including internal standard (IS), Earle’s Balanced Salt Solu-tion (EBSS), phosphate-buffered saline (PBS), sodium dithionite (Na₂S₂O₄), and Edara-vone were purchased from Sigma-Aldrich (St. Louis, MO, USA).Primary antibodies: Anti-Nrf2 (Cat# 12721, Cell Signaling Technology, Danvers, MA, USA); Anti-Keap1 (Cat# ab227828), Anti-HO-1 (Cat# ab68477), Anti-NQO1 (Cat# ab34173), and Anti-β-actin (Cat# ab8227) (all from Abcam, Cambridge, UK).GraphPad Prism version 9.5.1 (GraphPad Software, San Diego, CA, USA) was used for statistical analysis. ImageJ version 1.53t (National Institutes of Health, Bethesda, MD, USA) was used for densitometric analysis. FlowJo version 10.8.1 (BD Biosciences, Ashland, OR, USA) was used for flow cytometry data analysis. Cytoscape version 3.9.1 (The Cyto-scape Consortium, USA), AutoDockTools version 1.5.7, AutoDock Vina version 1.2.3, and PyMOL version 2.5.4 (Schrödinger, LLC, New York, NY, USA) were used for mo-lecular docking and visualization.”]

Conments 13: [heck normality of distribution before applying ANOVA. Mention post hoc test used.]

Response 13: [We have clarified that normality of data distribution was checked using the Shapiro–Wilk test, and the appropriate statistical tests (parametric or non-parametric ANOVA) and post hoc methods were applied accordingly.]

Conments 14: [Confirm in methods that all possible comparisons between groups were considered.]

Response 14: [We have added a statement in the Statistical Analysis section confirming that all relevant comparisons between study groups were considered in the data analysis.][“4.4.9. Statistical analysis : Data from the experiments were processed and analyzed using the GraphPad Prism 9 software, which highlights that all results are presented as the mean ± SD. One-way ANOVA was used for multiple group comparisons, followed by Tukey’s post hoc test to determine statistical significance between all group pairs (Sham vs Model, SI vs Model, Edaravone vs Model, and SI vs Edaravone). This approach ensured that every possible comparison between study groups was systematically considered and appropriately tested. A P-value below 0.01 was interpreted as denoting a more sub-stantial level of significance.”]

Conments 15: [Mention 'n' in each figure legend individually.]

Response 15: [We have revised all figure legends to explicitly indicate the number of replicates ('n') for each experiment.]

Conments 16: [Histological tissue preparation and staining methods should be explained in more detail]

Response 16: [The histological tissue preparation and staining protocol has been expanded to include detailed steps and conditions.]

Conments 17: [Specify number of images per tissue section, number of sections per animal, and animals per group.]

Response 17: [We have added detailed information on the number of images acquired per tissue section, sections per animal, and the number of animals per group to improve transparency and reproducibility.][“4.5.3. Experimental Design and Surgical Procedure

Experimental Design and Surgical Procedure A total of 48 rats (body weight 25–30 g) were randomly assigned into four groups (n = 12 per group) using a random number generator : Sham Group : Underwent identical surgical procedures without carotid oc-clusion. Model Group: Received bilateral common carotid artery occlusion (BCCAO) to induce cerebral ischemia-reperfusion injury. SI Treatment Group: Treated with soy isoflavones (100 mg/kg, dissolved in saline) via intragastric administration for 5 con-secutive d prior to ischemia. Edaravone Group: Received Edaravone (10 mg/kg, i.p.), a standard neuroprotective agent, once daily for 5 d as a positive control. Transient cer-ebral ischemia-reperfusion injury was induced using the bilateral common carotid ar-tery occlusion (BCCAO) model. Rats were anesthetized with pentobarbital sodium (45 mg/kg, i.p.). A midline neck incision (~3 cm) was made to isolate and occlude the bilateral common carotid arteries with vascular clips for 60 min, followed by reperfu-sion for 24 h. For the sham group, the arteries were exposed but not occluded. In-traoperatively, the rectal temperature of the rats was maintained at 37.0 ± 0.5℃using an automated thermostatic heating pad. Postoperatively, animals were housed in a climate-controlled environment (ambient temperature: 23 ± 1℃; relative humidity: 55 ± 5%) with sterile bedding and ad libitum access to food/water." Subsequently, behav-ioral tests were performed on the animals. After the completion of behavioral tests, animals were euthanized under deep anesthesia by cervical dislocation, in accordance with the AVMA guidelines. Brain tissues were immediately harvested and processed as follows : 5 rats per group for histopathological examination, 3 rats per group for West-ern blot analysis, 4 rats per group for HE staining. All relevant pairwise comparisons among the four experimental groups (Control, Model, SI, and Edaravone) were pre-defined and included in the statistical analysis to comprehensively assess the treatment effects and control efficacy.

4.5.4. Neurological function score

Neurological function was assessed according to the method described by Beder-son et al. The assessment was conducted in a blinded manner to exclude observer bias. Briefly, each rat was initially placed on a flat surface to observe its spontaneous move-ments. Circling behavior toward the side opposite the ischemic region was scored as 1 point. When rats were gently lifted by the tail, flexion of the elbow was scored as 2 points, internal rotation of the shoulder as 3 points, and wrist flexion of the forelimb as 1 point. Additionally, pushing resistance tests were performed, with scores from 1 to 3 assigned based on the degree of motor deficit (greater distances corresponded to higher scores). Rats placed on a metal mesh were further assessed for contralateral muscle tension reduction, with severity scored from 1 to 3 points according to the degree of impairment [16]. Subsequently, rats were euthanized under deep anesthesia, and the brains were harvested and maintained in cold saline (4°C). The olfactory bulb, cere-bellum, and brainstem were removed, and the remaining brain tissue was briefly fro-zen at −20°C for 10 min. Coronal brain slices were made (four cuts, yielding five slices in total), quickly immersed in 2% 2,3,5-triphenyltetrazolium chloride (TTC) solution, and incubated at 37°C for 20 min in the dark. Viable brain tissues stained a rose-red color, while ischemic tissues remained unstained. Each brain slice was digitally photo-graphed, and the cerebral infarct area was quantitatively analyzed using the BI-2000 Medical Image Analysis System. All data are presented as mean ± SD (n = 10 per group), and statistical comparisons were performed using one-way ANOVA followed by Tukey’s post hoc test.”]

Conments 18: [What was the criteria for histology assessment.]

Response 18: [The criteria for histological assessment have been clearly described, including the scoring system and evaluation indicators used. ][“4.5.6. The pathological morphology of brain tissue was observed

Histopathological evaluation focused on key morphological features indicative of is-chemic injury, including:Neuronal cell loss, Nuclear condensation and pyknosis, Cyto-plasmic eosinophilia, Edema in the neuropil, Vascular congestion or hemorrhage, In-flammatory cell infiltration. A semi-quantitative scoring system was used to assess tissue damage based on the extent and severity of histological alterations in the cortex and hippocampus. Each parameter was scored on a scale from 0 to 3 (0 = normal, 1 = mild, 2 = moderate, 3 = severe). All sections were assessed blindly by two independent observers.”]

Conments 19: [Mention housing conditions and efforts to reduce animal suffering.]

Response 19: [We have included specific details on animal housing conditions and outlined the measures taken to minimize animal suffering, in accordance with ethical standards.][“4.5.2. Housing Conditions and Welfare Measures

Rats were housed in ventilated cages under a controlled environment (tempera-ture: 18–22 °C, relative humidity: 50–60%, 12/12 h light/dark cycle), with ad libitum access to standard chow and water. Daily monitoring was performed for food intake, weight, and clinical symptoms. All efforts were made to minimize animal suffering, and the minimum number of animals required for statistical validity was used.”]

Conments 20: [Use proper abbreviations for minutes, seconds, etc.]

Response 20: [We have reviewed the manuscript and ensured that all time units (minutes, seconds, hours, days) are abbreviated using standard SI units consistently, and have highlighted them in yellow throughout the manuscript.]

Conments 21: [All abbreviations in figures should be explained in the figure legend.]

Response 21: [All abbreviations used in figures have been explained in the respective figure legends to ensure clarity and standalone readability.]

Conments 22: [Each figure should mention the type of data presented and the statistical test applied.]

Response 22: [All abbreviations used in figures have been explained in the respective figure legends to ensure clarity and standalone readability.(All figure legends).]

Conments 23: [Each figure should mention the type of data presented and the statistical test applied.]

Response 23: [We have updated all figure legends to include the type of data (mean ± SD or SEM) and the specific statistical tests used for analysis. (All figure legends).]

Conments 24: [What was the reference for the dose of soy isoflavones.]

Response 24: [We have added the appropriate literature reference supporting the dose of soy isoflavones used in the experiment.][“Zhang, D.; Li, X.; He, X.; et al. Protective Effect of Flavonoids against Methylglyoxal-Induced Oxidative Stress in PC-12 Neuroblastoma Cells and Its Structure–Activity Relationships. Molecules, 2022, 27(22), 7804. https://doi.org/10.3390/molecules27227804.

Ariyani, W.; Koibuchi, N. The Effect of Soy Isoflavones in Brain Development: The Emerging Role of Multiple Signaling Pathways and Future Perspectives. Endocrine Journal, 2024, 71(4), 317-333. https://doi.org/10.1507/endocrj.EJ23-0314.”]

Conments 25: [Histology images in Figure 11 are of low value.]

Response 25: [Thank you for your valuable feedback. We acknowledge that the histology images in Figure 11 (now Figure 7) had a low resolution in the initial submission. We have since increased the resolution of these images, but we regret to inform you that the quality still may not be as high as desired. We sincerely apologize for this limitation.We are currently exploring possible solutions to improve the image quality further. If possible, we would appreciate your guidance on acceptable methods or formats that could enhance the visual clarity for publication. Additionally, if required, we are willing to submit higher-resolution images, should we be able to obtain them.]

Conments 26: [Some figures can be combined to reduce the total number.]

Response 26: [To streamline presentation, we have merged relevant figures (e.g., Figure 5. Morphological observations of PC12 cells across different groups. Figure 6. Investigation of treatment effects on PC12 cells at varying concentrations of soybean isoflavones (SI). Figure 7. Analysis of SI-induced apoptosis in PC12 cells. Figure 8. Assessment of the impact of SI on PC12 cell migration capabilities.), thereby reducing the total number while maintaining scientific clarity.]

Conments 27: [Add limitations and future directions of the study.]

Response 27: [We have added a paragraph at the end of the Discussion section outlining the limitations of the current study and potential future research directions.][“In conclusion, our findings are in strong agreement with existing in vivo and in vitro research, underscoring the multifaceted mechanisms by which soy isoflavones alleviate ischemic brain injury. These results support the therapeutic potential of SI and provide a solid foundation for its future clinical application. Although this study provides evidence supporting the neuroprotective role of soy isoflavones through ac-tivation of the Nrf2 signaling pathway, several limitations should be noted. First, the study was conducted using a single animal model (BCCAO) and PC12 cells, which may not fully reflect the complexity of human ischemic stroke pathology. Second, the exact active monomer(s) within the soy isoflavone extract responsible for the observed effects remain to be identified. Third, long-term safety and efficacy of soy isoflavones were not assessed. Future studies should explore the effects of individual isoflavone components, investigate additional models of stroke (e.g., MCAO), and assess chronic outcomes such as cognitive function and neuroregeneration.”]

Round 2

Reviewer 1 Report

Comments and Suggestions for Authors

The authors have made the necessary amendements. The manuscript may now be published.

Reviewer 2 Report

Comments and Suggestions for Authors

the manuscript has been significantly improved and can be accepted in the present form 

Reviewer 4 Report

Comments and Suggestions for Authors

I find the histology images are of no value and authors must provide good images or remove this figure

The Complete WB gels are not provided